# PerTurboID, a targeted in situ method reveals the impact of kinase deletion on its local protein environment in the cytoadhesion complex of malaria-causing parasites

**Heledd Davies[1], Hugo Belda[1], Malgorzata Broncel[1], Jill Dalimot[1], Moritz Treeck[1,2]\***

[1]Signalling in Apicomplexan Parasites Laboratory, The Francis Crick Institute, London, United Kingdom; [2]Cell Biology of Host-Pathogen Interaction Laboratory, Gulbenkian Institute of Science, Oeiras, Portugal

**Abstract** Reverse genetics is key to understanding protein function, but the mechanistic connection between a gene of interest and the observed phenotype is not always clear. Here we describe the use of proximity labeling using TurboID and site-specific quantification of biotinylated peptides to measure changes to the local protein environment of selected targets upon perturbation. We apply this technique, which we call PerTurboID, to understand how the *Plasmodium falciparum*-exported kinase, FIKK4.1, regulates the function of the major virulence factor of the malaria-causing parasite, PfEMP1. We generated independent TurboID fusions of two proteins that are predicted substrates of FIKK4.1 in a FIKK4.1 conditional KO parasite line. Comparing the abundance of site-specific biotinylated peptides between wildtype and kinase deletion lines reveals the differential accessibility of proteins to biotinylation, indicating changes to localization, protein–protein interactions, or protein structure which are mediated by FIKK4.1 activity. We further show that FIKK4.1 is likely the only FIKK kinase that controls surface levels of PfEMP1, but not other surface antigens, on the infected red blood cell under standard culture conditions. We believe PerTurboID is broadly applicable to study the impact of genetic or environmental perturbation on a selected cellular niche.

**\*For correspondence:**
moritz.treeck@crick.ac.uk;
mtreeck@igc.pt

**Competing interest:** The authors declare that no competing interests exist.

## Editor's evaluation

This important study combines conditional mutagenesis with proximity labeling to evaluate alterations in a sub-cellular proteome upon a perturbing event. The approach is applied to the deletion of a kinase involved in trafficking of adhesins to the malaria parasite-infected erythrocyte surface and the evidence supporting the conclusions is compelling. The work will be of broad interest to cell biologists and biochemists.

## Introduction

Malaria is caused by parasites of the genus *Plasmodium*. Of at least six *Plasmodium* species that infect humans, *Plasmodium falciparum* is responsible for the vast majority of the approximately 600,000 annual malaria-associated deaths (***World Health Organization, 2022***). The particular severity of *P. falciparum* malaria is largely due to the cytoadhesion of the infected red blood cell (iRBC) to the endothelium of the host vascular system. Cytoadhesion is mediated by the *Plasmodium falciparum* Erythrocyte Membrane Protein 1 (PfEMP1), a surface protein family of ~60 variants, which bind to different

**eLife digest** Enzymes known as protein kinases regulate a huge variety of biological processes inside cells by attaching small tags known as phosphate groups onto specific locations on certain proteins. For example, the parasite that causes malaria infections in humans and great apes, injects a protein kinase called FIKK4.1 into certain cells in its host. This enzyme then adds phosphate groups to various parasite and host proteins that, in turn, causes them to form a large group of proteins (known as the cytoadhesion complex) to protect the parasite from being cleared by the hosts' immune defences. However, it remains unclear how and where the complex forms, and how the parasite regulates it.

Proximity labelling is a well-established method that allows researchers to label and identify proteins that are near to a protein of interest. To investigate how the FIKK4.1 enzyme alters host cells to make the cytoadhesion complex, Davies et al. combined proximity labelling with methods that disturb the normal state of cells at a specific timepoint during development. The team used this new approach – named PerTurboID – to identify the proteins surrounding three components in the cytoadhesion complex. This made it possible to create a map of proteins that FIKK4.1 is likely to modify to build and control the cytoadhesion complex.

Further experiments examined what happened to these surrounding proteins when FIKK4.1 was inactivated. This revealed that some protein targets of FIKK4.1 become either more or less accessible to other enzymes that attach a molecule known as biotin to proteins. This could be a result of structural changes in the cytoadhesion complex that are normally regulated by the FIKK4.1 kinase.

In the future, PerTurboID may be useful to study how genetics or environmental changes affect other groups of proteins within specific environments inside cells, such as protein complexes required for DNA replication or cell division, or assembly of temporal structures required for cell movement.

endothelial receptors including ICAM1, CD36, EPCR, and Chondroitin Sulfate-A (CSA) (*Baruch et al., 1996*; *Salanti et al., 2004*). Cytoadhesion-induced accumulation of iRBCs in the vasculature of various organs, particularly the brain and placenta, can lead to fatal outcomes for the patient.

PfEMP1 is trafficked through the parasite's secretory pathway into the parasitophorous vacuole, then exported by the PTEX translocon into the RBC. Here it is believed to be transported via protein complexes called J-dots into Maurer's clefts (*Külzer et al., 2012*; *Behl et al., 2019*; *Kriek et al., 2003*), which are membranous sorting stations established by the parasite (*Przyborski et al., 2003*). Around 20 hr post-infection (hpi), PfEMP1 is translocated onto the RBC surface and assembled into parasite-derived knob protrusions (*Crabb et al., 1997*; *Watermeyer et al., 2016*). This process is poorly understood, and it is unclear whether other surface proteins, such as the immunomodulatory RIFIN family proteins, are trafficked by the same machinery (*Goel et al., 2015*; *Saito et al., 2017*). Coinciding with PfEMP1 surface translocation is an increase in RBC rigidity, likely through re-arrangement of the RBC spectrin cytoskeleton (*Rug et al., 2014*).

Parasite proteins which are exported into the host cell mediate the assembly of knobs on the iRBC cytoskeleton and the trafficking of PfEMP1 via Maurer's clefts. Knob-associated histidine-rich protein (KAHRP) is a key component of the knobs (*Rug et al., 2006*), and a subset of proteins from the *Plasmodium* helical interspersed subtelomeric (PHIST) family interact with the PfEMP1 cytosolic domain and are thought to anchor it to the erythrocyte cytoskeleton (*Oberli et al., 2014*; *Yang et al., 2020*; *Proellocks et al., 2014*). Seven proteins have been specifically designated as PfEMP1-trafficking proteins (PTPs) due to the loss or reduction in surface PfEMP1 presentation upon their deletion (*Maier et al., 2008*; *Carmo et al., 2022*).

We have previously shown that a parasite-exported kinase belonging to the FIKK family, FIKK4.1, is important for efficient surface translocation of the PfEMP1 variant Var2CSA and its adhesion to its receptor CSA, as well as for modulating iRBC rigidity (*Davies et al., 2020*). Using a DiCre-mediated conditional FIKK4.1 knock-out (*Jones et al., 2016*; *Collins et al., 2013*) and quantitative phosphoproteome analysis, we identified 66 parasite and RBC proteins that are phosphorylated in an FIKK4.1-dependent manner. KAHRP, several PTPs, and PHIST proteins are among these likely substrates of FIKK4.1, as well as proteins involved in nutrient acquisition (*Ito et al., 2017*) and several proteins of unknown function. Phosphorylation likely alters the dynamics of several of these proteins to ultimately influence PfEMP1 presentation, while other substrates may be non-functional or play a role in an

unrelated phenotype. This is true for kinase–substrate relationships in general (*Ochoa et al., 2020*), and therefore mutation of individual phosphorylation sites is rarely informative for understanding the complex interactions at play.

Here, we developed a method that relies on proximity labeling using TurboID fusion proteins and mass spectrometry to identify changes to protein accessibility in a targeted manner. TurboID is a biotin-ligase that, if fused to a bait, biotinylates lysines on proteins within an ~10 nm radius (*Branon et al., 2018*). By using a protocol designed for enrichment and detection of biotinylated peptides rather than whole proteins (*Udeshi et al., 2017*; *Kim et al., 2018*), we identify protein domains accessible to biotinylation. We generated FIKK4.1 conditional KO lines harboring TurboID fusions of two FIKK4.1 substrates, KAHRP and PTP4, allowing us to monitor protein dynamics in both these proteins' vicinity in the presence and absence of FIKK4.1. Using this novel combination of cell perturbation and TurboID, which we call PerTurboID, we gain insights into protein localization, topology, protein–protein interactions, and post-translational modifications upon FIKK4.1 deletion. We show that among the 17 exported FIKK kinases, modulation of PfEMP1 surface translocation is unique to FIKK4.1, and that other surface proteins such as RIFINs are not affected by FIKK4.1 deletion, indicating a specific pathway for the surface translocation of this key virulence determinant.

## Results
### Identification of the FIKK4.1 subcellular niche using TurboID
To measure the effect of FIKK4.1-dependent phosphorylation of its substrates using proximity labeling, we first identified appropriate bait proteins that are likely direct substrates of FIKK4.1 and present in its vicinity. To do so, we generated a C-terminally tagged TurboID fusion of FIKK4.1 using CRIPSR/Cas9 (*Birnbaum et al., 2017*). Clones were selected, PCR validated (*Figure 1—figure supplement 1*), and the specific labeling of proteins in the presence of biotin was validated by western blot (*Figure 1A*). Substantial biotinylation could be observed in FIKK4.1-TurboID transgenic parasites (hereafter called FIKK4.1T) but not in NF54 wildtype controls, where only minimal background biotinylation is present. We visualized the localization of biotinylated proteins in transgenic parasites in the absence and presence of biotin using streptavidin conjugated to a fluorophore (*Figure 1B*). In biotin-free media, some background labeling was observed within the parasite, while the addition of biotin leads to substantial biotinylation of proteins within the parasitophorous vacuole and the red blood cell compartment, particularly at the iRBC periphery where FIKK4.1 is localized (*Davies et al., 2020*). To ensure that the FIKK4.1-TurboID fusion does not impact FIKK4.1 function, we performed PfEMP1 surface translocation assays by flow cytometry. The PfEMP1 variant Var2CSA was trafficked normally onto the iRBC surface in the absence of biotin and after a 4 hr biotin pulse (*Figure 1C* and *Figure 1—figure supplement 2*). Collectively these data show that the FIKK4.1T fusion is expressed, active, and does not compromise FIKK4.1 function.

We identified proteins in the vicinity of FIKK4.1 using in situ labeling with biotin, followed by the pulldown of proteins by neutravidin-coated beads, and label-free quantification of trypsin-digested proteins using mass-spectrometry (*Figure 1D* and *Supplementary file 1*). Wildtype (NF54) and FIKK4.1T parasites were pulsed with biotin between 16–20 hpi or 40–44 hpi in triplicate then lysed in 1% SDS buffer. We chose the above timepoints to capture changes occurring as PfEMP1 is assembled onto the iRBC surface starting at ~16 hpi. In total, we quantified 101 parasite proteins and 39 human proteins that showed at least a 1.5-log2 fold enrichment in the FIKK4.1T samples compared to NF54 at either timepoint (*Supplementary file 2*). 62.3% of the 101 identified parasite proteins are predicted to be exported, a significant enrichment compared to the *Plasmodium* proteome which only contains ~8% exported proteins (*Aurrecoechea et al., 2009*). The parasite proteins with the strongest enrichment include FIKK4.1 itself and three PfEMP1 trafficking proteins (PTP4, PTP2, and PTP3), which are also predicted FIKK4.1 substrates (*Davies et al., 2020*; *Figure 1E*). The most abundant human proteins include members of the 4.1R complex, suggesting that FIKK4.1 specifically interacts with this complex (*Figure 1F*). Notably, we found substantial labeling of alpha-synuclein, which is mainly associated with vesicle trafficking in neurons (*Lashuel et al., 2013*), with no known role in RBCs.

There is a clear enrichment of exported *Plasmodium* proteins associating with FIKK4.1 at later stages (20% early versus 66% late) (*Figure 1G*). This is likely because FIKK4.1 is expressed and exported to the RBC periphery in early ring stages before most other exported proteins are present

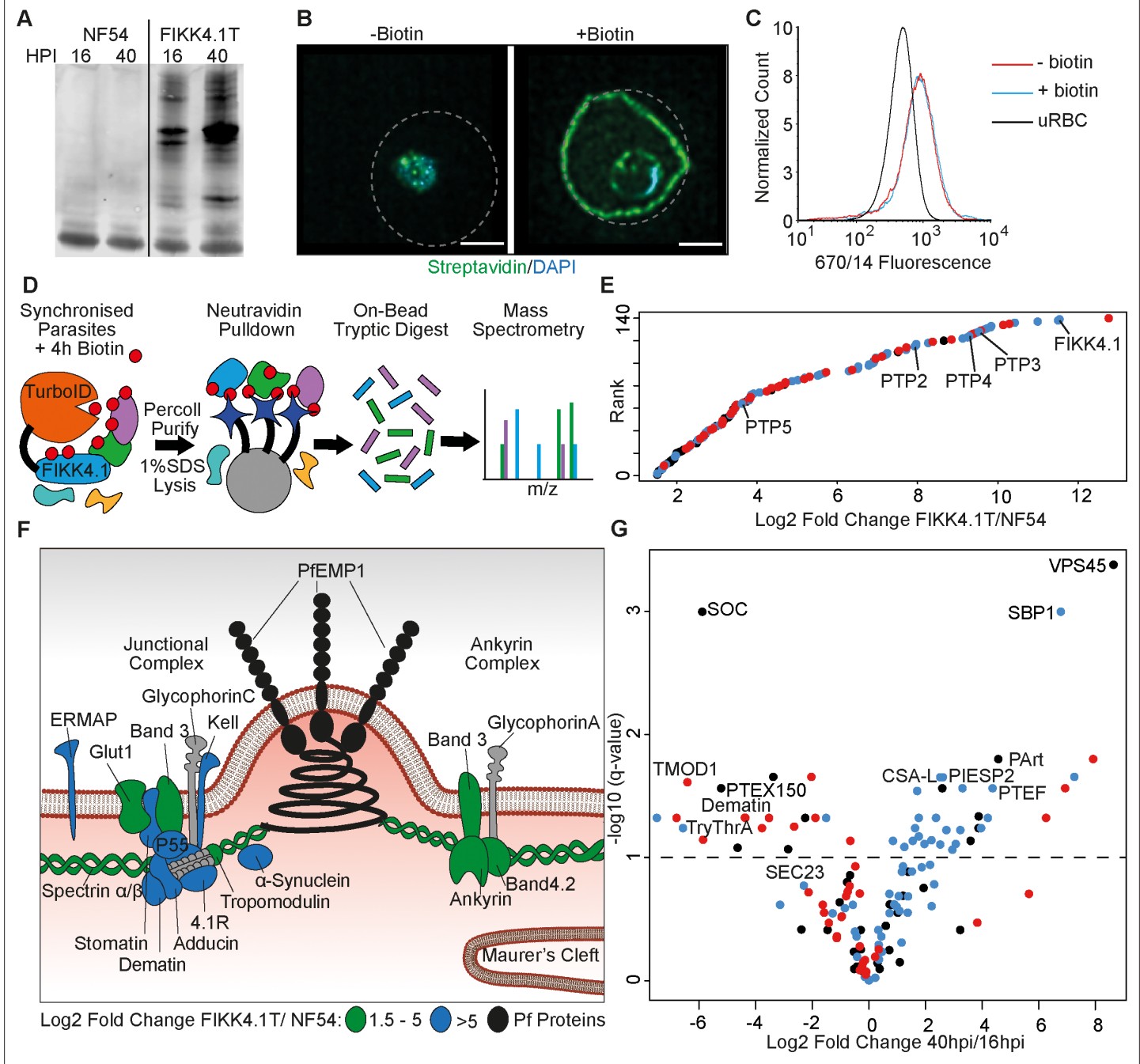

**Figure 1.** Proximity-dependent protein labeling of the FIKK4.1 subcellular niche. (**A**) Western blot of parental NF54 and the FIKK4.1-TurboID (FIKK4.1T) fusion line probed with streptavidin conjugated to a fluorophore. Parasites were incubated with biotin either between 16–20 hr post-infection (hpi) or 40–44 hpi. (**B**) Structured illumination microscopy of FIKK4.1T fusion line incubated with or without biotin, and probed with streptavidin-fluorophore (green), and DAPI (blue). Dotted lines represent the approximate position of the RBC membrane according to brightfield images (see *Figure 1—figure supplement 3*). Scale bar: 2 µm. (**C**) Flow cytometric analysis of FIKK4.1T fusion lines using anti-Var2CSA antibodies. (**D**) Schematic of the classical TurboID protocol. TurboID bait fusion parasites are incubated with biotin (red), purified, then lysed. Biotinylated proteins are enriched, digested, and peptides quantified using mass spectrometry. (**E**) Ranked plot of all proteins quantified above NF54 background (x-axis – log2 fold change FIKK4.1T/ NF54). Red, human proteins; blue, exported parasite proteins; black, non-exported parasite proteins. (**F**) Schematic of a knob structure at the infected red blood cell (iRBC) cytoskeleton. Human proteins with log2 fold change FIKK4.1T/ NF54 from 1.5 to 5 are highlighted in green, and the most abundant proteins with values >5 are in blue. Parasite proteins are in black, and other RBC proteins are in gray (**G**) Volcano plot depicting the log2 fold change in abundance between early (16–20 hpi) and late (40–44 hpi) biotin pulses with FIKK4.1T parasites. Red, human proteins; blue, exported *P. falciparum* proteins; black, non-exported *P. falciparum* proteins. Selected proteins are labeled.

*Figure 1 continued on next page*

*Figure 1 continued*

The online version of this article includes the following source data and figure supplement(s) for figure 1:

**Source data 1.** Unedited blot for *Figure 1A*.

**Figure supplement 1.** PCRs confirming correct genome modification of the FIKK4.1 locus.

**Figure supplement 1—source data 1.** Original data for *Figure 1—figure supplement 1*.

**Figure supplement 2.** Verification of Var2CSA surface translocation in FIKK4.1-TurboID transgenic lines.

**Figure supplement 3.** Structured illumination microscopy of FIKK4.1-TurboID lines with brightfield images included.

(*Davies et al., 2020*), increasing the relative proportion of human proteins at the early stage. We also detect proteins which FIKK4.1 presumably encounters as it is trafficked into the RBC, such as the protein export translocon component PTEX150 which is located in the parasitophorous vacuole membrane surrounding the parasite (*de Koning-Ward et al., 2009*). This protein was ~37-fold more biotinylated at the 16 hr timepoint compared to the 40 hr timepoint. In later stages it is likely that the majority of FIKK4.1 protein is located within the RBC, rather than within the trafficking pathway, and proteins biotinylated at this point are more likely to operate within the same subcellular niche.

## Generation and validation of PerTurboID parasite lines to monitor the local protein environment upon FIKK4.1 deletion

Of the 101 proteins identified in the proximity of FIKK4.1, we selected two candidates to use as sensors of the local protein environment upon FIKK4.1 deletion: PfEMP1-trafficking protein 4 (PTP4) and KAHRP. KAHRP is a component of the knob structures, whereas PTP4 was previously observed in punctate structures in the RBC cytosol (*Rug et al., 2006*; *Jonsdottir et al., 2021*). Both PTP4 and KAHRP have been identified as likely direct targets of FIKK4.1 (*Davies et al., 2020*) and are found in the TurboID dataset at approximately equal amounts at both timepoints, indicating an early interaction with the kinase. Both proteins are important for the correct presentation of PfEMP1 in knobs on the erythrocyte surface (*Maier et al., 2008*; *Horrocks et al., 2005*).

A FIKK4.1 conditional KO line was previously generated and cloned using rapamycin-induced DiCre-mediated excision of the kinase domain. A TurboID cassette was integrated at the C-terminus of PTP4 or KAHRP in this FIKK4.1 conditional KO line using CRISPR/Cas9 and selection-linked integration (*Figure 2A*; *Birnbaum et al., 2017*; *Lee et al., 2019*; *Knuepfer et al., 2017*). Parasites were cloned by serial dilution and plaque formation (*Thomas et al., 2016*), and correct integration of selected clones was validated by PCR (*Figure 2—figure supplement 1*). Both transgenic parasite lines expressed the TurboID fusion and retained the ability to excise the FIKK4.1 kinase domain upon treatment with rapamycin (*Figure 2—figure supplement 1*). No difference in PfEMP1 surface translocation was observed between the transgenic parasite lines and FIKK4.1 cKO parental parasites or after a 4 hr biotin pulse (*Figure 2B* and *Figure 2—figure supplement 2*), indicating that PfEMP1 transport is not affected by the TurboID tag. Western blots of biotin-treated parasites showed that the KAHRP-TurboID fusion had a different and substantially stronger banding pattern than PTP4-TurboID (*Figure 2C* and *Figure 2—figure supplement 3*). RAP treatment did not result in obvious changes beyond potential differences in loading, indicating that no prominent proteins in the vicinity of KAHRP and PTP4 are completely lost upon FIKK4.1 deletion. Visualization of KAHRP and FIKK4.1 using structured illumination microscopy (SIM) revealed a partial co-localization at the erythrocyte periphery, although FIKK4.1 appears to be positioned more toward the lumen of the cell (*Figure 2D* and *Figure 2—figure supplement 4*). PTP4 also appeared to localize mainly at the iRBC periphery, with some puncta within the iRBC cytosol. PTP4 co-localized only marginally with FIKK4.1 and was generally closer to the lumen of the red blood cell (*Figure 2E* and *Figure 2—figure supplement 4*). Treatment with biotin resulted in streptavidin-fluorophore labeling predominantly around the periphery of the erythrocyte for both fusion lines (*Figure 2F and G* and *Figure 2—figure supplement 4*), with puncta visible when focusing on the top of the cell (*Figure 2—figure supplement 5*). Despite little overlap by IFA, western blot analysis shows that both PTP4 and KAHRP TurboID fusions were able to biotinylate FIKK4.1, resulting in its pulldown by neutravidin-coated beads, suggesting transient co-localization (*Figure 2—figure supplement 3*).

We then tested the optimal biotin labeling time to obtain a sufficient signal from the KAHRP and PTP4 TurboID fusion proteins (hereafter referred to as KAHRP-T and PTP4-T) for measurement by mass spectrometry. Western blot analysis of lysates from parasites pulsed for 2 hr, 4 hr, and 6 hr with

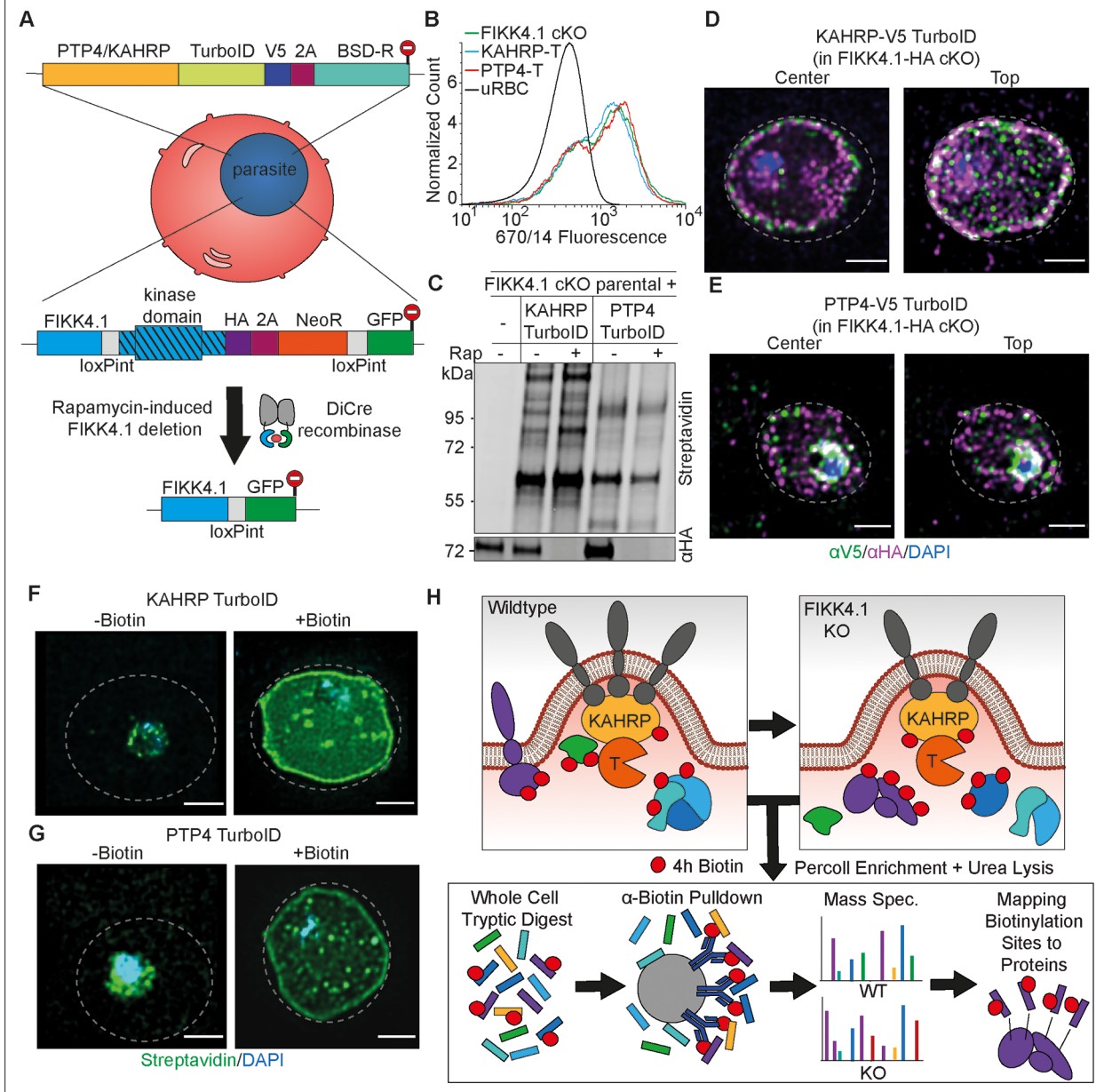

**Figure 2.** Design and validation of KAHRP and PTP4-TurboID tagging in a FIKK4.1 cKO line for PerTurboID. (**A**) Schematic of the generation of double transgenic parasite lines containing a loxP-flanked FIKK4.1 kinase domain for conditional KO and KAHRP/PTP4-TurboID fusions. (**B**) Surface translocation of Var2CSA in KAHRP/PTP4-TurboID and parental lines analyzed by flow cytometry using anti-Var2CSA antibodies. (**C**) Western blot of FIKK4.1 cKO parental line and KAHRP/PTP4 TurboID-fusion parasites, with full-length FIKK4.1 (-RAP) or the kinase domain deleted (+RAP). The blot was probed with streptavidin-fluorophore or anti-HA antibodies targeting FIKK4.1-HA, confirming excision and biotinylation. (**D, E**) Structured illumination microscopy of KAHRP/PTP4:TurboID fusion lines probed with anti-HA (FIKK4.1, magenta), anti-V5 (PTP4 or KAHRP, green), and DAPI, blue. Views from the top and center of the cell are shown. (**F, G**) TurboID transgenic parasites incubated with or without biotin and probed with streptavidin-fluorophore (green) and DAPI (blue). Dotted lines represent the approximate position of the RBC membrane according to bright field images (see *Figure 2—figure supplement 5*). Scale bar: 2 µm. (**H**) Schematic depicting the PerTurboID protocol. FIKK4.1 cKO induces hypothetical changes to proteins in the vicinity of KAHRP/PTP4, leading to differences in the accessibility of lysines to biotinylation. Parasites are purified, lysed, then digested. Biotinylated peptides are enriched, then eluted and quantified by mass spectrometry. Biotinylated peptides are mapped onto the protein, indicating protein domains accessible to the TurboID biotin ligase. T, TurboID tag.

The online version of this article includes the following source data and figure supplement(s) for figure 2:

**Source data 1.** Unedited blot for *Figure 2C*.

**Figure supplement 1.** PCRs confirming correct genome modification of the PTP4 and KAHRP locus.

*Figure 2 continued on next page*

**Figure supplement 1—source data 1.** Unedited DNA gels for *Figure 2—figure supplement 1*.

**Figure supplement 3.** Western blot analysis of KAHRP and PTP4-TurboID fusion lines.

**Figure supplement 3—source data 1.** Unedited blots for *Figure 2—figure supplement 3*.

biotin suggested a 4 hr pulse was sufficient to achieve a clear biotinylation signal (*Figure 2—figure supplement 3*). We chose a 4 hr pulse between 36 and 40 hpi for mass spectrometry analysis as the effects of FIKK4.1 deletion on PfEMP1 surface exposure are observable at this stage (*Davies et al., 2020*). To measure the local protein environment of PTP4 and KAHRP and how it changes upon FIKK4.1 deletion, we performed proximity labeling of both DMSO (control) and RAP-treated (FIKK4.1 KO) parasite lines in triplicate. To observe the differential accessibility of protein domains and not just whole proteins to the PTP4- and KAHRP-T fusions, we chose a workflow based on methods by *Kim et al., 2018* and *Udeshi et al., 2017*, where the full proteome is first digested and biotinylated peptides subsequently enriched by anti-biotin antibodies rather than neutravidin beads. The weaker affinity of biotin to anti-biotin antibodies compared to neutravidin allows for biotinylated peptides to be eluted and quantified by label-free mass spectrometry (*Figure 2H*). Identified peptides are then mapped onto proteins.

## Mapping biotinylated sites provides insights into protein structure, interactions, topology, and localization

In total, we obtained 3560 peptides across the datasets, of which 2272 were biotinylated (*Supplementary file 3*). All non-biotinylated peptides were excluded from the dataset, eliminating the background signal often observed using the classical BioID/TurboID protocol. The biotinylation sites, as well as co-occurring phosphorylation sites, were then mapped to proteins (see *Supplementary file 4* for site-specific analysis and *Supplementary file 5* for peptide-specific analysis). In the KAHRP-T dataset, we identified 195 proteins, while 150 were found in the PTP4-T dataset. To identify shared and unique proteins in the local protein environment of FIKK4.1, PTP4, and KAHRP, we mapped all interactions in a network (*Figure 3—figure supplement 1* and *Supplementary file 6*). All three proteins are observed in each other's datasets, suggesting that they co-localize at least transiently. Interactions with several other host and parasite exported proteins from Maurer's clefts, J-dots, and the iRBC periphery are also observed in all three datasets, suggesting that these subcellular organelles partially co-localize. The export translocon component, PTEX150, was the only protein not predicted to be exported which appeared in all three datasets, highlighting the specificity of the TurboID labeling for proteins in the proximity of the three exported bait proteins. The identification of PTEX150, which resides in the parasitophorous vacuole, indicates that some of the shared targets may originate from the common trafficking pathway. However, since the KAHRP- and PTP4-T parasites were pulsed for 4 hr at late stages, it is likely that the majority of the observed interactions are the result of partial co-localization at the proteins' final locations.

The site-specific proteomics workflow we used allows us to localize the biotinylated peptides within protein sequences. From such data, one can infer information about the accessibility of lysines to biotinylation. For example, biotinylation on either side of a transmembrane domain provides an insight into protein topology, and we predict that biotinylation would be largely excluded from protein–protein interaction interfaces and within structurally buried regions. Movement of a domain away from the TurboID fusion protein can also result in reduced biotinylation of protein sequences. To investigate to what extent the localized biotinylation sites correspond to accessible domains of proteins and with predicted protein topology, we looked at some examples in detail. The structure of the RHOPH complex (PDB: 7KIY) (*Schureck et al., 2021*; see also *Ho et al., 2021*), composed of RHOPH3, Clag3.1, and RHOPH2, illustrates several facets of information that can be captured by our peptide-centric TurboID dataset. This parasite-exported protein complex is inserted into the RBC membrane where it is thought to form the plasmodial surface anion channel (PSAC) which is essential for the acquisition of nutrients for parasite growth (*Ito et al., 2017*; *Desai et al., 2000*). The structure

represents the soluble form of the complex, likely representing its state as it is trafficked through the iRBC cytosol before insertion into the membrane (*Schureck et al., 2021*; *Figure 3A*). Disordered regions not resolved in the electron microscopy structure have been filled in with the AlphaFold-predicted domains (*Jumper et al., 2021*).

Most biotinylation sites on the RHOPH complex are localized in the disordered domains not resolved by electron microscopy (*Figure 3A*). This is expected as disordered domains are more likely to be surface exposed. Interestingly, however, biotinylation sites 1321 and 1323 within the component Clag3.1 are partially buried within the complex, suggesting that a state exists where these sites are more exposed to the TurboID-fusion proteins, potentially upon insertion into the iRBC membrane. RHOPH3 is biotinylated on both sides of its transmembrane domain, indicating that it was likely biotinylated in the soluble state represented by the structure, where the transmembrane domain is sequestered within the core of the complex during trafficking and both biotinylated sites are exposed to the cytosol (*Schureck et al., 2021*). Localization of the TurboID fusion proteins at the erythrocyte periphery likely allows both the soluble and membrane-inserted structural states of the complex to be captured.

While exported proteins have a propensity towards disordered sequences (*Guy et al., 2015*), some exported protein families contain conserved folded domains. Twenty-nine PHIST proteins were identified in our dataset (*Supplementary file 7*). These proteins contain an alpha-helical PRESAN domain, a subset of which have been shown to interact with the cytosolic domain of PfEMP1 (*Oberli et al., 2014*; *Yang et al., 2020*; *Proellocks et al., 2014*). Mapping the location of either KAHRP- or PTP4-T-mediated biotinylation on these proteins demonstrates a substantial reduction in biotinylation within the PRESAN domain itself as opposed to most of the surrounding sequences, despite no reduction in the number of lysines in this domain (*Figure 3B*). It is therefore likely that lysines in this domain are occluded from biotinylation by protein–protein interactions, potentially with PfEMP1.

As neither KAHRP nor PTP4 contain transmembrane domains, their TurboID tags likely remain within the RBC cytosol and can only biotinylate protein domains in this compartment or within the trafficking pathway. This gives us added insight into the topology of exported proteins with transmembrane domains which are inserted into the membrane of Maurer's clefts or the iRBC. For example, we observe biotinylation only at the N-terminus of Maurer's cleft protein SEMP1, meaning that it is likely that its single transmembrane domain is inserted into the membrane with the N-terminus exposed to the RBC cytosol. Conversely, several other Maurer's cleft proteins with N-terminal transmembrane domains appear to be inserted into the Maurer's clefts in the opposite orientation, with the C-terminus exposed to the TurboID tags (*Figure 3C*). The biotinylation of the surface proteins Var2CSA and RIFIN shows contrasting patterns, with Var2CSA biotinylated only on the cytosolic C-terminal ATS domain, while RIFIN is biotinylated on the N-terminal extracellular domain (*Bachmann et al., 2015*). It is likely that this RIFIN biotinylation occurs during the trafficking process where this domain may be at some point accessible to the RBC cytosol, or within the parasitophorous vacuole.

While many of the biotinylated proteins were found in both KAHRP and PTP4 TurboID experiments, we also found several that were specific or highly enriched for one of the two fusion proteins. In *Figure 3D*, we have highlighted 35 proteins for which almost all peptides are more biotinylated either by KAHRP or PTP4-TurboID fusions, likely reflecting their positions in the cell relative to the two proteins (*Supplementary file 8*). Among the 16 proteins more biotinylated by KAHRP-TurboID are several proteins known to localize to the erythrocyte periphery such as Clag3.1, LYMP, MESA, GARP, and PHISTb proteins (*Oberli et al., 2014*; *Proellocks et al., 2014*; *Ito et al., 2017*; *Davies et al., 2016*; *Tarr et al., 2014*; *Coppel et al., 1988*), as well as human alpha-synuclein. Proteins more closely associated with PTP4 include parasite chaperones HSP70x, PF3D7_0113700 (HSP40), and PF3D7_0501100 (HSP40), as well as host chaperone CCT8. This agrees with previous immunoprecipitation experiments suggesting that PTP4 interacts with J-dots which are enriched in chaperones (*Külzer et al., 2012*; *Behl et al., 2019*; *Jonsdottir et al., 2021*). Surprisingly, we found that Var2CSA PfEMP1 is more biotinylated by PTP4 than KAHRP-T, despite the fact that KAHRP is thought to directly associate with the PfEMP1 intracellular domain (*Ganguly et al., 2015*).

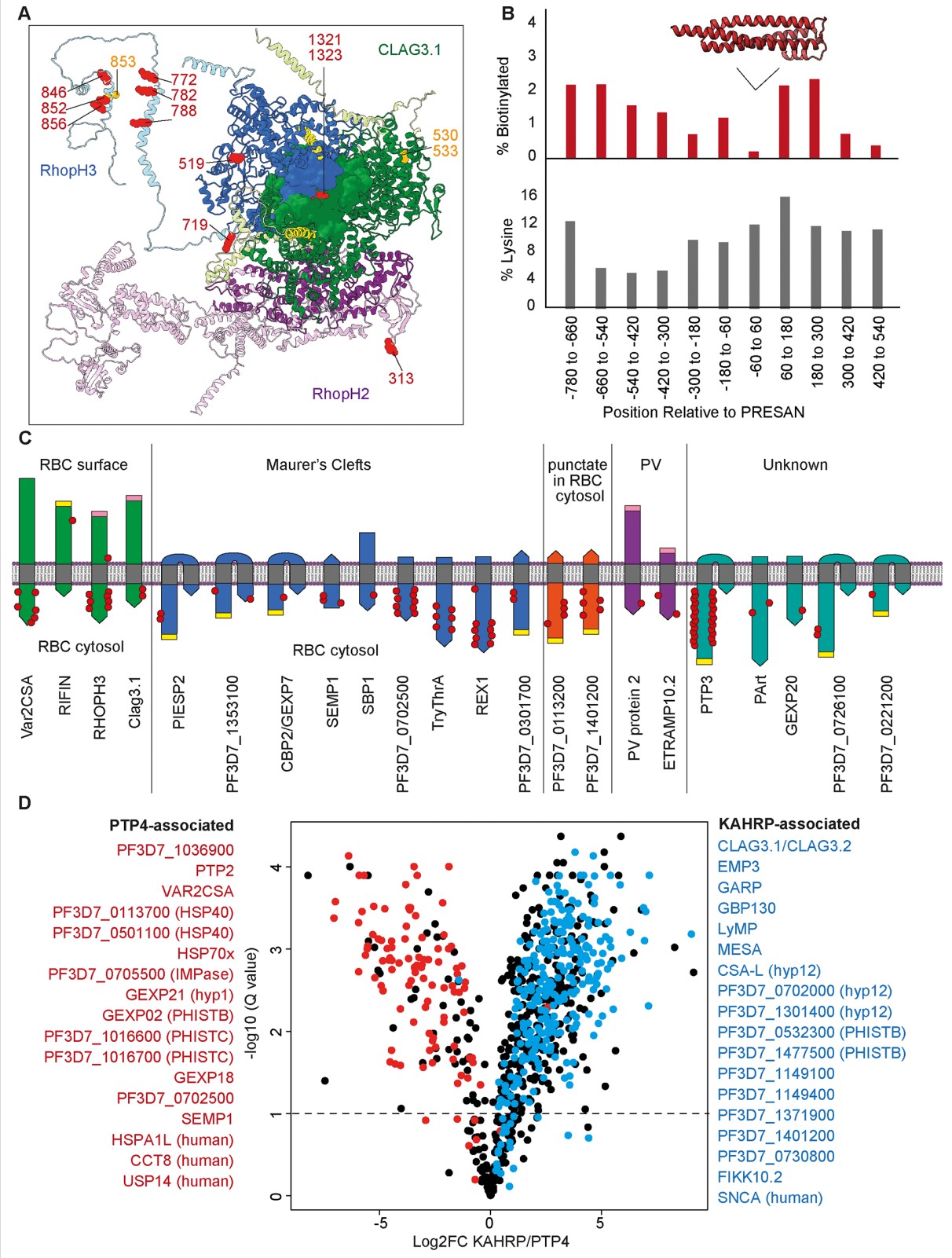

**Figure 3.** Insights into protein structure, interactions, topology, and localization from KAHRP and PTP4 site-specific TurboID. (**A**) Cryo-electron microscopy structure (PDB: 7KIY) of the RHOPH complex (RHOPH2, purple; RHOPH3, blue; Clag3.1, green) (*Schureck et al., 2021*). Unresolved regions have been substituted with the AlphaFold-predicted structures and rotated to prevent clashes (lighter shades) (*Jumper et al., 2021*). Lysines biotinylated by either KAHRP or PTP4-TurboID are represented as red side-chain spheres. Residues that are either more or less phosphorylated upon

*Figure 3 continued on next page*

*Figure 3 continued*

FIKK4.1 deletion are in orange. Transmembrane domains are in yellow. Image created in Chimera (*Pettersen et al., 2004*). (**B**) A representation of the position of biotinylation of lysines across all 29 *Plasmodium* helical interspersed subtelomeric (PHIST) proteins in the dataset. Positions (x-axis) are relative to the predicted center of the ~120 aa PRESAN domain which spans positions –60 to 60 (PRESAN position predicted by PFAM via PlasmoDB; *Amos et al., 2022*). Top panel: y-axis represents the percentage of all lysine residues which are biotinylated in either KAHRP or PTP4 datasets. Bottom panel: percentage of lysines in primary amino acid sequence. A drop in biotinylation is observed within the PRESAN domain itself, with no reduction in the proportion of lysine residues. (**C**) Schematic depicting transmembrane domain (gray) proteins observed in the dataset, with the orientation based either on the literature (for PfEMP1, RIFIN, and SBP1) or predicted from the position of biotinylation sites. Some MC and punctate proteins may be trafficked in chaperoned complexes before insertion into a membrane at a later stage – we depict this hypothetical 'membrane-inserted' form. N-termini are flat, C-termini are pointed. PEXEL motifs, which are cleaved during export, are in yellow. Predicted TM domains and signal peptides upstream of the PEXEL motif are removed. Cleaved signal peptides with no PEXEL motif are in pink. Proteins are grouped according to published subcellular localization. Where proteins occupy more than one subcellular position (e.g. PIESP2 has been found in MC and on the surface), they are placed in one group only. The approximate positions of biotinylated residues are depicted with red dots. Proteins are not to scale. (**D**) Volcano plot depicting the log2 fold change in biotinylated peptide abundance between KAHRP and PTP4-TurboID fusion parasites. Proteins for which almost all peptides are more abundant in either PTP4 (red) or KAHRP (blue) datasets are highlighted. Peptides originating from KAHRP or PTP4 themselves have been removed for clarity. y-axis (-log10 q-value, N = 3).

The online version of this article includes the following figure supplement(s) for figure 3:

**Figure supplement 1.** Network analysis of proteins observed in FIKK4.1, KAHRP, and PTP4 TurboID datasets.

## PerTurboID identifies changes in protein accessibility to biotinylation upon FIKK4.1 deletion

To identify changes in the local protein environment of PTP4 and KAHRP upon deletion of the FIKK4.1 kinase domain, we compared the abundance of biotinylated peptides in FIKK4.1 cKO DMSO vs. RAP-treated conditions at 36–40 hpi (*Supplementary files 4 and 5*). A barcode format, similar to that used for representing limited proteolysis-mass spectrometry (LiP-MS) data (*Cappelletti et al., 2021*), was used to illustrate the change in accessibility to biotinylation of lysine residues in proteins observed in the dataset upon FIKK4.1 deletion (*Figure 4A*). Each row represents a lysine in a given protein which is gray if not observed or colored according to the log2 fold change in intensity upon FIKK4.1 deletion. A barcode for both KAHRP and PTP4 datasets was generated for all proteins where at least one site displayed log2 fold change >1 or <–1 upon FIKK4.1 deletion (*Figure 4A* and *Figure 4—figure supplement 1*). Some biotinylated peptides were also phosphorylated; fold changes for these are calculated separately to their non-phosphorylated counterparts and depicted with their own barcodes for KAHRP in *Figure 4A* and all other proteins in *Figure 4A* and *Figure 4—figure supplement 1*. The position of FIKK4.1-dependent phosphorylated residues observed previously is indicated by red or blue dots. Since the kinase domain of FIKK4.1 is deleted upon RAP treatment, we found that in both KAHRP and PTP4-TurboID RAP-treated parasite lines there was a loss of biotinylation in this region, as expected (*Figure 4A*). An increase in biotinylation upstream of the kinase domain was observed, likely as the remaining N-terminus of the protein is more exposed to biotinylation due to loss of the folded kinase domain. This serves as a useful control for FIKK4.1 excision and illustrates the capability of the PerTurboID protocol to reproducibly detect changes in protein accessibility in both PTP4 and KAHRP-T lines.

KAHRP and PTP4 themselves showed substantial, but different, changes in the biotinylation pattern upon FIKK4.1 deletion (*Figure 4A*). An increase in biotinylation upon FIKK4.1 deletion is observed in the PTP4 C-terminus, while the N-terminus appears mostly unchanged. For KAHRP, the most substantial changes occur around position 351–364 (denoted by asterisks in *Figure 4A*). This section encompasses predicted FIKK4.1 phosphorylation sites observed in our previous phosphoproteomics experiments (*Davies et al., 2020*). In the PerTurboID datasets, both phosphorylated and non-phosphorylated versions of the biotinylated peptides encompassing this section of KAHRP are detected. As expected, the phosphorylated peptides become less abundant upon FIKK4.1 deletion due to the loss of kinase activity. This leads to an increase in the relative abundance of the non-phosphorylated counterparts of the same sequence (see *Figure 4B* for schematic). A similar effect is seen for the FIKK4.1 substrate acyl CoA synthetase 7 (ACS7) around position 876, whereas the nutrient channel component RHOPH3 is more phosphorylated at position 853 upon FIKK4.1 deletion, shifting the ratio of phosphorylated to non-phosphorylated peptides in the opposite direction (see *Supplementary file 9* for non-phosphorylated/phosphorylated peptide ratios). These observations

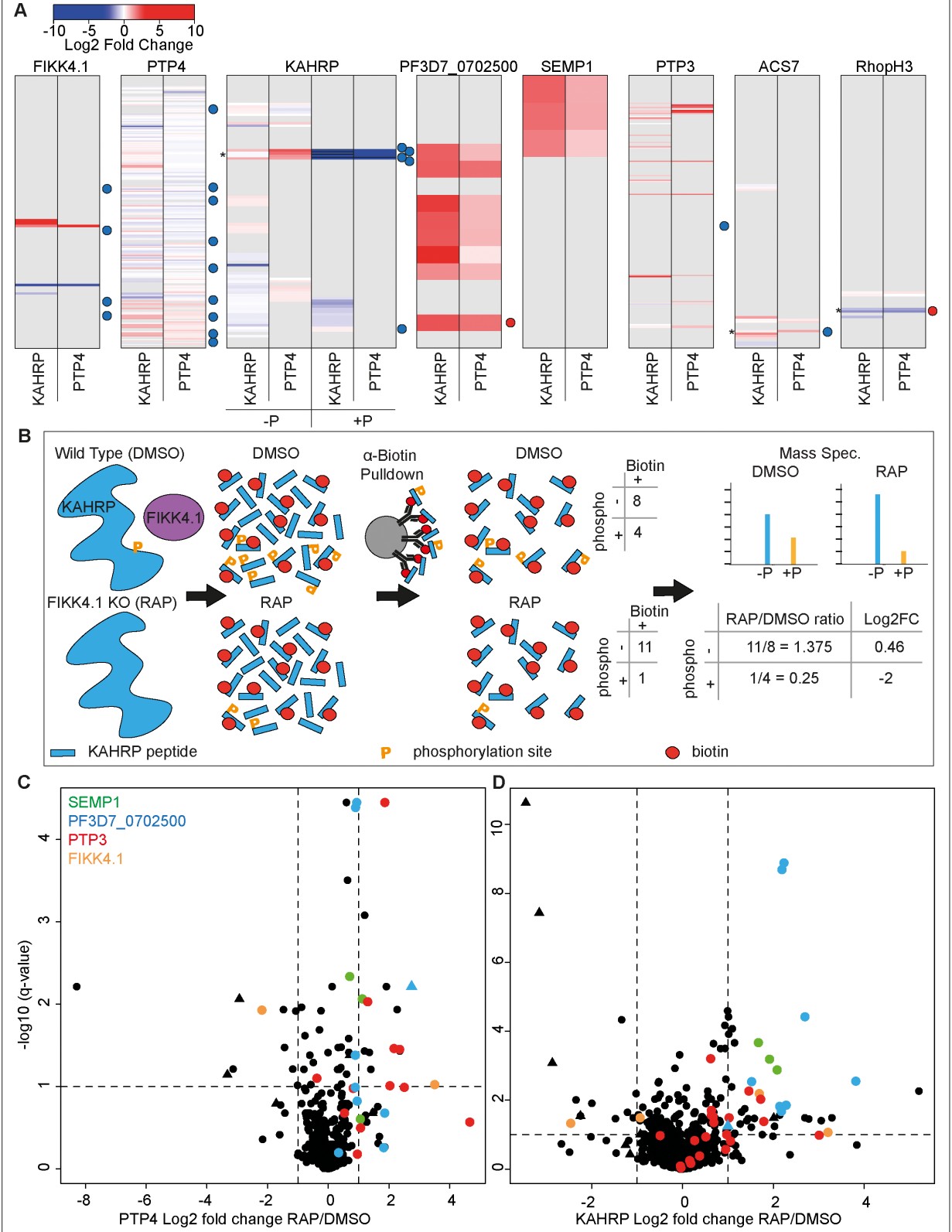

**Figure 4.** Changes in site-specific biotinylation observed upon FIKK4.1 deletion. (**A**) Barcodes representing selected proteins from the PerTurboID dataset. Each row represents a lysine, with the N-terminus of the protein at the top. Lysines not biotinylated in either the KAHRP (left) or PTP4 (right) dataset are in gray. Biotinylated lysines are colored according to the log2 fold change in abundance upon FIKK4.1 deletion (RAP/DMSO), with more abundant biotinylated peptides in red and less abundant in blue. For KAHRP, peptides which are both phosphorylated and biotinylated are shown in

*Figure 4 continued on next page*

*Figure 4 continued*

the right-hand panels (P+) while those biotinylated only are on the left (P-). Phosphorylated peptides for all other proteins are shown in *Figure 4—figure supplement 1*. Dots to the right of the barcodes represent the approximate position of phosphorylation sites on the protein which were previously shown to be affected by FIKK4.1 deletion (blue, less phosphorylated; red, more phosphorylated upon FIKK4.1 deletion). Asterisks denote positions of interest (351–364 of KAHRP, 876 of ACS7, and 853 of RHOPH3). (**B**) Schematic demonstrating how PerTurboID can detect changes in protein phosphorylation, using KAHRP as an example. Peptides are represented as blue bars with either one or two modifications: phosphorylation (orange 'P') or biotin (red circle). Biotinylated peptides that are also phosphorylated are quantified separately from peptides that are only biotinylated. Loss of phosphorylated + biotinylated peptides upon FIKK4.1 deletion therefore leads to an increase in the biotinylated-only peptide and a positive log2 fold change. (**C, D**) Volcano plots showing the log2 fold change in biotinylated site abundance between FIKK4.1 wildtype and kinase-deleted conditions (RAP/DMSO) for PTP4-TurboID (**B**) and KAHRP-TurboID (**C**) fusion parasites. Biotinylated sites from four parasite proteins, SEMP1, PTP3, Pf3D7_0702500, and FIKK4.1, are highlighted. Sites which are also phosphorylated are represented by triangles.

The online version of this article includes the following figure supplement(s) for figure 4:

**Figure supplement 1.** Barcodes representing changes to lysine biotinylation upon FIKK4.1 KO on proteins in the PerTurboID dataset.

**Figure supplement 2.** Peptide-level volcano plots.

are supported by our earlier phosphoproteomics experiments (*Davies et al., 2020*) and demonstrate the ability of PerTurboID to detect changes in post-translational modifications without specifically enriching for modified peptides.

Across all proteins, a similar number of peptides increased and decreased in biotinylation intensity (see *Figure 4C and D* for site-level volcano plots and *Figure 4—figure supplement 2* for peptide-level plots). While KAHRP and PTP4 display changes to protein accessibility on a local level, other proteins show substantial changes to all biotinylated peptides. Most notably, PTP3, SEMP1, and PF3D7_0702500 each show an increase in biotinylation upon deletion of FIKK4.1 at almost all sites in both datasets, suggesting a more substantial change to the localization or the accessibility of the proteins (*Figure 4C and D*). SEMP1 and PF3D7_0702500 co-localize at the Maurer's clefts (*Dietz et al., 2014*), while PTP3 appears to localize to the iRBC cytosol (*Maier et al., 2008*), although the presence of two predicted transmembrane domains suggests it may also associate with membranes. Of note, one site (position 2694) on Var2CSA PfEMP1 itself exhibited significantly increased biotinylation upon FIKK4.1 deletion, suggesting that it becomes more exposed. Other nearby biotinylation sites on Var2CSA do not change; the alterations leading to the defect in surface presentation may therefore be very subtle. Collectively these results indicate that FIKK4.1 activity leads to altered protein dynamics in the local environment of the cytoadhesion complex, including some previously identified PfEMP1 trafficking proteins.

## FIKK4.1 is the only FIKK kinase important for trafficking multiple PfEMP1 variants but not RIFINs

Since other exported FIKK kinases localize to the periphery of the RBC or the Maurer's clefts (*Davies et al., 2020*; *Nunes et al., 2010*), we wanted to test whether any of the FIKK kinases in addition to FIKK4.1 play a role in the surface translocation of PfEMP1. To do so, we performed PfEMP1 surface translocation assays with all conditional FIKK kinase deletion lines we previously generated (*Davies et al., 2020*). Parasites were treated with rapamycin to delete the kinase domain and Var2CSA PfEMP1 surface translocation was assessed by flow cytometry at ~40 hpi in the subsequent cycle. Only FIKK4.1 showed a significant reduction in anti-Var2CSA antibody binding under the conditions tested here (*Figure 5A*). This reduction in antibody binding is unlikely caused by misfolding of Var2CSA PfEMP1 as a reduced signal is observed upon FIKK4.1 deletion using four different antibodies raised against various domains of the protein (antibodies kindly gifted by Lars Hviid, University of Copenhagen) (*Barfod et al., 2007*; *Figure 5B*). This suggests that the defect arises from reduced PfEMP1 exposure on the iRBC surface rather than changes in the conformation of the extracellular region, as this would be less likely to affect all domains equally.

Since Var2CSA is one of ~60 PfEMP1 variants, with only one expressed in a parasite at any given time, we wanted to test whether FIKK4.1 is important for trafficking other PfEMP1 variants. To do so, we selected for binding to other endothelial cell receptors by panning over CD36-expressing CHO cells (*Lubiana et al., 2020*). Treatment of these FIKK4.1 cKO parasites with rapamycin results in a 51.5 ± 13% reduction in cytoadhesion under flow to CHO cells expressing CD36 (*Figure 5C*). This is similar to the 54% reduction in binding of Var2CSA-expressing parasites to CSA-expressing HBEC-5i cells

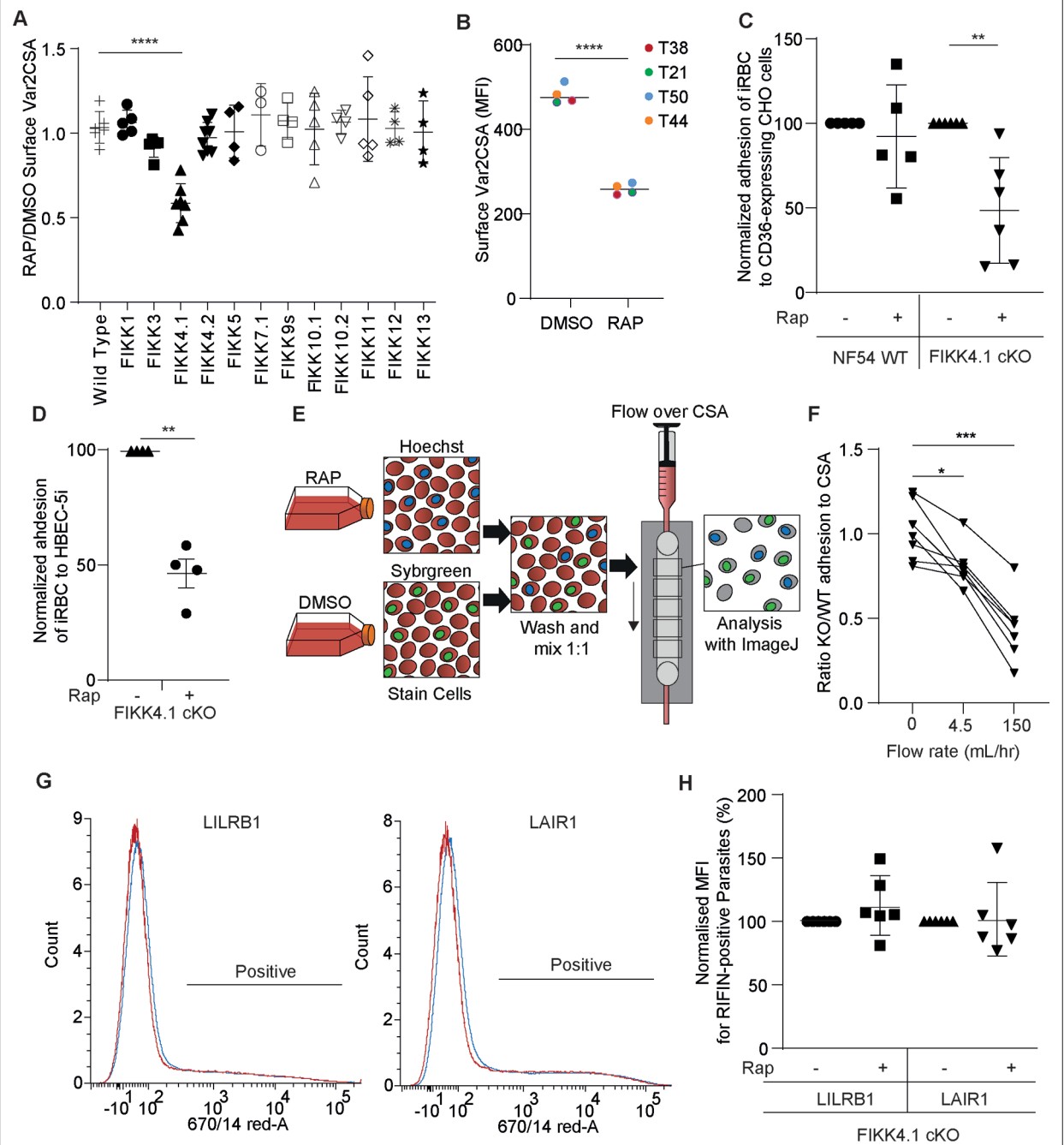

**Figure 5.** FIKK4.1 is the only FIKK kinase to modulate surface translocation of PfEMP1, but not of RIFINs. (**A**) Var2CSA surface translocation analysis by flow cytometry using anti-Var2CSA antibodies (*Srivastava et al., 2011*). y-axis, ratio of median fluorescence intensity of Var2CSA-positive cells between RAP and DMSO-treated parasites lines. x-axis, conditional kinase domain deletion lines for the indicated FIKK kinase. 'FIKK 9s' represents the deletion of seven FIKK kinases on chromosome 9. Analyzed by one-way ANOVA with Dunnet's multiple-comparison test. FIKK4.1 vs. wildtype – adjusted p-value<0.0001, n = 3–6. (**B**) Flow cytometry analysis of the surface presentation of Var2CSA in WT (DMSO) or FIKK4.1 KO parasites (RAP) using monoclonal antibodies directed towards different extracellular domains of the protein. y-axis, median fluorescence intensity. Paired *t*-test p-value<0.0001. (**C**) Adhesion assays with NF54 parental or FIKK4.1 cKO parasites to CD36-expressing CHO cells under flow, with or without RAP treatment, normalized to the no RAP control. Analyzed by one-way ANOVA with Sidak's multiple-comparison adjustment. FIKK4.1 – Rap vs. +Rap adjusted p-value=0.0014. (**D**) Binding of iRBC to CSA-expressing HBEC-5i cells under flow. Numbers are normalized to the DMSO control. *t*-test p-value=0.0033. (**E**) Schematic depicting the set-up for semi-automated analysis of adhesion to CSA under flow. Dyed wildtype or KO parasites are premixed and flowed over CSA-coated channels. Images are taken along the channel and the ratio of wildtype to KO quantified by ImageJ at different flow speeds. (**F**) FIKK4.1 KO/WT adhesion under flow using the method depicted in (**E**). x-axis, flow rate in mL/hr. Images for 'before' measurements were taken with a separate uncoated channel with no flow. y-axis, ratio of WT/KO parasites at different flow speeds. Analysis by paired one-way ANOVA

*Figure 5 continued on next page*

*Figure 5 continued*

with corrected for multiple comparisons by the Sidak method. p-values: before vs. 4.5 mL/hr = 0.0228, before vs. 150 mL/hr = 0.0008. (**G**) Representative flow cytometry histograms of RIFIN presentation on FIKK4.1 WT (red) or KO (blue) infected red blood cell (iRBC). iRBCs were incubated with LILRB1-Fc or LAIR1-Fc fusion proteins, followed by anti-human Fc-APC and SYBR green staining. Parasites positive for LILRB1 or LAIR1-binding RIFINs are indicated. (**H**) Histograms from (**G**) are quantified, with the median fluorescence intensity of positive parasites normalized to the untreated control. No significant difference was observed.

(*Figure 5D*), suggesting that FIKK4.1 likely has a general role in modulating PfEMP1 surface translocation, and our previous observations were not limited to Var2CSA PfEMP1.

To further dissect how cytoadhesion is affected by FIKK4.1 deletion, we investigated how mimicking blood flow affects the avidity of binding. Wildtype and FIKK4.1 cKO lines were divided into two, treated with either rapamycin or DMSO, then stained with different DNA dyes and premixed before flowing over CSA-coated channels. Exposing both lines to the same conditions allowed us to control for differences in CSA coating, fluctuations in flow rates, and initial parasitemia (*Figure 5E*). The quantification of relative binding was automated by fluorescent image analysis in ImageJ (*Schindelin et al., 2012*). Parasites were initially allowed to flow over CSA-coated channels at 4.5 mL/hr for 10 min, a slow rate which allows cells to roll along the surface of the channel. A difference in binding between the FIKK4.1 KO and wildtype parasites is already apparent at this point, suggesting that some cells lose their ability to bind CSA entirely (*Figure 5F*). By gradually increasing the flow rate to 150 mL/hr, a much stronger reduction in binding was observed in the FIKK4.1 KO line, indicating that the avidity of binding is also affected.

RIFIN family proteins are also exposed on the iRBC surface and several have been shown to bind immune cell receptors LILRB1 and LAIR1, leading to a downregulation of the host immune response (*Saito et al., 2017*). We previously found that one RIFIN was differentially phosphorylated upon FIKK4.1 deletion, suggesting that the kinase may also be important for its trafficking (*Davies et al., 2020*). To test this, we used flow cytometry analysis of iRBC binding to Fc-tagged LILRB1 and LAIR1 to test the surface translocation of RIFINs. No difference was observed upon FIKK4.1 deletion (*Figure 5G and H*), suggesting that FIKK4.1 is restricted to the trafficking of PfEMP1, indicating that there are independent mechanisms for the surface translocation of RIFINs and potentially other surface proteins.

## Discussion

We have combined conditional gene deletion with a peptide-centric TurboID proximity labeling approach, which we have termed PerTurboID, to better understand the complex downstream effects of gene deletion on a subcellular niche. We believe that this powerful method is widely applicable to study the effects of protein deletion or inactivation in a local environment. Here, we used PerTurboID to study the functions of FIKK4.1, an important regulator of efficient PfEMP1 surface translocation, a key factor for *P. falciparum* virulence (*Davies et al., 2020*). We identified several proteins that undergo accessibility changes upon FIKK4.1 deletion, which may be connected to the PfEMP1 surface translocation defect. KAHRP and PTP4 were selected as molecular surveillance tools in the FIKK4.1 cKO line as they are highly phosphorylated in a FIKK4.1-dependent manner and are important for the presentation of PfEMP1 in knobs (*Rug et al., 2006*; *Maier et al., 2008*). Changes to the biotinylation pattern were observed for both KAHRP and PTP4 upon FIKK4.1 deletion; however, the most abundant accessibility changes were observed in proteins which are not highly phosphorylated by FIKK4.1. Neither SEMP1 nor PF3D7_0702500 are predicted substrates of the kinase, and PTP3 is phosphorylated at just one site. This suggests that PerTurboID can identify downstream changes in localization or folding of proteins that could not be predicted to be affected by kinase inactivation using phosphoproteomics. While SEMP1 deletion does not affect PfEMP1 trafficking and PF3D7_0702500 is uncharacterized (*Dietz et al., 2014*), PTP3 deletion completely eliminates PfEMP1 on the RBC surface (*Maier et al., 2008*). Given the known function of both PTP3 and PTP4 for PfEMP1 surface translocation, it is possible that FIKK4.1 acts via one or both of these proteins to facilitate efficient PfEMP1 surface translocation, and thus cytoadhesion.

Whether the observed changes to Maurer's cleft proteins and the PSAC nutrient channel composed of the RHOPH complex contribute to PfEMP1 surface translocation is unclear. The reduction in PfEMP1 at the parasite surface and its potential accumulation along the trafficking pathway may

have a downstream effect on the mobility and accessibility of surrounding proteins. Although FIKK4.1 deletion affects the phosphorylation of RHOPH complex components RHOPH3 and Clag3.1 (*Davies et al., 2020*), it has no apparent effect on parasite viability, suggesting that FIKK4.1 is not responsible for the function of this essential complex under the conditions tested. Immunoprecipitation experiments suggest that RHOPH proteins interact with J-dots and Maurer's cleft proteins during trafficking, and that SEMP1 and PF3D7_0702500 co-localize at Maurer's clefts but also interact with J-dot proteins, with SEMP1 also translocated to the RBC membrane in late stages (*Jonsdottir et al., 2021*; *Dietz et al., 2014*). Our observed changes suggest that FIKK4.1 deletion may interfere with connections between trafficking vesicles, J-dots, and Maurer's clefts, perhaps by modulating PTP3 or PTP4.

We found that RIFIN transport was not impaired by FIKK4.1 deletion, whereas other PfEMP1 variants were similarly affected, suggesting a general PfEMP1-trafficking pathway distinct from the trafficking of other surface proteins. While no other FIKK kinase was important for controlling the total amount of PfEMP1 on the surface under the conditions tested here, they may function redundantly or modulate the efficiency of PfEMP1 trafficking or its conformation at earlier timepoints or under other conditions. For example, fever-like temperatures lead to an earlier increase in surface exposure of PfEMP1 (*Udomsangpetch et al., 2002*), which could be regulated by other FIKK kinases. They could also modulate cytoadhesion in other ways; FIKK4.2 deletion leads to aberrant knob structures, for example (*Kats et al., 2014*). Our optimized flow adhesion system could facilitate the analysis of cytoadhesion with other FIKK deletion lines and PfEMP1-receptor pairs.

We illustrate the potential for PerTurboID to measure changes to a multitude of protein characteristics in situ, including localization, topology, protein–protein interactions, structure, and post-translational modifications. The detection of several differentially phosphorylated sites upon FIKK4.1 deletion is validated by our previous phosphoproteome (*Davies et al., 2020*), and our topology analyses agree with previous experiments using selective permeabilization to study surface and Maurer's cleft protein orientation (*Bachmann et al., 2015*; *Blisnick et al., 2000*; *Saridaki et al., 2009*). The detection of proteins specific to either KAHRP or PTP4 demonstrates the precision of TurboID for detecting subtle changes not visible by western blot or immunofluorescence. Despite both probes appearing to be peripherally localized in the RBC, KAHRP-associated proteins included mainly cytoskeletal components, whereas PTP4 associated with chaperones known to form J-dots, to which PTP4 has previously been localized. Interestingly, human alpha-synuclein was highly associated with KAHRP, indicating that it may reside at the erythrocyte cytoskeleton (*Barbour et al., 2008*). By using multiple different probes in different localizations and at different timepoints, PerTurboID can be used to further dissect the temporal regulation of protein trafficking.

Some caveats are important to consider: the length of biotin pulses should be restricted to avoid functional effects from excessive protein biotinylation. TurboID fusion proteins have the potential to capture random interactions. To gain confidence in direct vs. indirect interactions, the recently published XL-BioID could provide higher temporal resolution by freezing protein complexes in time through chemical crosslinking (*Geoghegan et al., 2022*). For essential proteins, a non-functional probe protein localizing to the niche of interest could be TurboID-tagged to avoid indirect effects on parasite growth. Proteins high in lysines are likely to be over-represented in the data, while small proteins with fewer lysines may be missed. Fortunately, the alternative peptide-based processing method overcomes the main limitation of the traditional proximity labeling workflow: differentiating true interactions (biotinylated proteins) from the high background binding to beads. Here, non-biotinylated peptides can be identified and removed from the dataset, removing the necessity of including a wildtype or no-biotin control and considerably simplifying sample preparation and data analysis. This advantage is exemplified by the increased specificity for exported proteins in the KAHRP and PTP4-TurboID peptide-level dataset relative to the FIKK4.1-TurboID protein-level data.

PerTurboID can therefore complement other global methods probing protein accessibility such as limited proteolysis with mass spectrometry (LiP-MS) (*Cappelletti et al., 2021*) or thermal protein profiling (*Savitski et al., 2014*), with the targeted approach potentially allowing for a finer dissection of a specific subcellular niche and overcoming difficulties in capturing low-abundance proteins in whole-cell preparations. While we apply this technique to cell perturbation by gene deletion, it has the potential to investigate many other changes to the local protein environment, including to measure the effects of drug treatment and external stressors, as well as in measuring temporal changes such as membrane protein trafficking across time.

# Materials and methods

## In vitro maintenance and synchronization of parasites

*P. falciparum* parasites were cultured at 37°C in RPMI-1640 medium supplemented with 0.5% w/v AlbuMAX II, 25 mM HEPES, 11 mM glucose, 0.1 mM hypoxanthine, 21.4 mM sodium bicarbonate, and 12.5 µg/mL gentamycin (preformulated by Thermo Fisher Scientific). For TurboID experiments, RPMI-1640 medium was substituted with biotin-free RPMI-1640 medium (custom media, Thermo Fisher Scientific) and supplemented as above. The parasites were grown in blood provided through the UK National Blood and Transfusion service under parasite gas atmosphere (90% $N_2$, 5% $CO_2$, and 5% $O_2$). Parasites were synchronized by enriching schizont-stage parasites on a cushion of 65% Percoll (GE Healthcare), allowing them to invade uninfected RBC while shaking for 1–4 hr, then removing uninvaded schizonts with a second Percoll enrichment step.

## Generation of parasite lines

All primers used for the construction of plasmids and verifying correct modification of the parasite genome are included in *Supplementary file 10*. For the generation of FIKK4.1-TurboID, the Cas9 Homology repair plasmid including the TurboID-V5 sequence was ordered from Integrated DNA Technologies in a pMK-RQ plasmid. This plasmid was XhoI digested to linearize and precipitated together with a PDC2 plasmid containing a Cas9 cassette, the tracR RNA and guide, and an hDHFRuFCU-resistance cassette for positive selection with WR99210 and negative selection by Ancotil (*Knuepfer et al., 2017*). The plasmids were transfected into egressing schizonts using an Amaxa electroporator and Lonza 4D-Nucleofector kit with P3 primary cell buffer. Parasites were selected with 2.5 nM WR99210 at days 1–5 following transfection.

All FIKK cKO lines were generated as described previously (*Davies et al., 2020*) using selection-linked integration or Cas9 to insert loxP-introns to flank the kinase domain. The FIKK4.1 cKO parasites generated through this method were resistant to WR99210 and G418. To endogenously tag KAHRP and PTP4 with TurboID in the FIKK4.1 cKO line, a T2A skip peptide followed by a blasticidin resistance cassette was introduced into the locus. 5′ homology regions, recodonized KAHRP/PTP4, TurboID-V5, T2a-blasticidinR, and 3′ homology regions were PCR-amplified and Gibson-assembled into a pMK-RQ plasmid, which was linearized and transfected together with the Cas9-guide plasmids. Drug pressure with 2 µg/mL blasticidin was applied 4 d after transfection and sustained until resistant parasites emerged. All parasite lines were cloned by serial dilution and plaque formation (*Thomas et al., 2016*) and verified by PCR.

## Microscopy

Coverslips were coated with ConA, then washed with PBS before adding 20 µL of parasite culture in PBS. Parasites were allowed to settle for 15 min before washing gently with PBS 3×. Parasites were fixed in 4% PFA and 0.0075% GA for 30 min, ensuring the coverslips were not allowed to dry. Parasites were washed then permeabilized in 0.1% Tx-100 for 10 min, then washed with PBS and blocked for 30 min with 1% BSA in PBS. The primary antibody was added for 1 hr, before washing 3× with PBS, and adding the secondary for 1 hr. After five washes with PBS, a drop of ProLong Gold Antifade Mountant with DAPI (Thermo Fisher Scientific) was added to each coverslip before mounting onto a slide for imaging.

For detecting FIKK4.1-HA, cells were probed with rat anti-HA (clone 3F10, Roche; 1:1000), followed by 1:10,000 anti-rat Alexa 594 secondary. For detecting biotinylated proteins, cells were probed with 1:5000 streptavidin-Alexa 488. For detecting KAHRP or PTP4 V5, cells were probed with either rabbit anti-V5 (MA5-32053, Invitrogen; 1:200), followed by 1:10,000 anti-rabbit Alexa 488, or with mouse anti-V5 (ab27671, Abcam, 1:200) followed by 1:10,000 anti-mouse Alexa 488 (all secondary antibodies from Life Technologies).

Parasites were imaged with a VisiTech iSIM microscope, and images were deconvolved and analyzed in Huygens Professional software.

## Western blot

Percoll-enriched schizonts were resuspended in PBS, solubilized in protein loading buffer, and denatured at 95°C for 10 min. Parasite extracts were resolved by SDS-PAGE gel electrophoresis, then transferred to Transblot turbo mini-size nitrocellulose membranes (Bio-Rad) and blocked for 1 hr

in 3% BSA in PBS at 4°C. For detecting FIKK4.1-HA, membranes were probed with rat anti-HA (clone 3F10, Roche; 1:1000) followed by 1:20,000 IRDye 680RD goat anti-rat IgG secondary antibody (LI-COR Biosciences). For detecting KAHRP and PTP4-V5, membranes were probed with rabbit anti-V5 (MA5-32053, Invitrogen; 1:200), followed by 1:20,000 IRDye 800CW donkey anti-rabbit IgG secondary antibody (LI-COR Biosciences). For detecting biotinylated proteins, blots were probed with fluorophore-conjugated streptavidin IRDye 800CW (LI-COR Biosciences) at a 1:1000 dilution. All antibody incubations were performed in 3% BSA in PBS for 1–2 hr. Blots were washed 3× in PBS with 0.2% Tween 20. Blots were visualized using the Odyssey infrared imaging system (LI-COR Biosciences).

## Proximity labeling experiments

### TurboID: Protein-level sample processing

#### Parasite enrichment

NF54 wildtype parasites, used as controls, and FIKK4.1-TurboID parasites were tightly synchronized to a 4 hr window using Percoll. Parasites were grown in triplicate in 200 mL biotin-free complete medium and 2 mL of blood to at least 10% parasitemia, with each replicate cultured in blood from a different donor. Then, 4 hr biotin pulses were performed at two different timepoints (16–20 hpi or 40–44 hpi) by adding 100 μM D-biotin (bioAPE) to the cultures in a shaking incubator. After biotinylation, iRBCs were washed five times in 50 mL biotin-free complete medium to remove biotin. IRBCs treated with biotin between 16 and 20 hpi were returned to culture in biotin-free complete medium and allowed to reach late schizont stage (44 hpi), then harvested by Percoll and washed five times with 5 mL PBS. Purified parasites were lysed in 1% SDS in PBS containing protease inhibitors (cOmplete, Roche) on ice for 30 min. Samples were further solubilized by sonication with a microtip sonicator on ice for three rounds of 30 se at a 30% duty cycle. Lysates were then clarified by centrifugation at 21,000 × $g$ for 30 min at 4°C. Samples were diluted 3× in PBS to reduce SDS to <0.4%, and the protein concentration of the samples was calculated using a BCA protein assay kit (Pierce).

#### Pulldown

Acetylated high-capacity Neutravidin agarose beads (40 μL slurry/1 mg sample) were equilibrated by washing 3 × 3 min with five bead volumes of 0.3% SDS/PBS, centrifuging at 1500 × $g$ for 2 min between washes. The lysates (3 mg of protein/sample) were added to equilibrated beads and rotated at room temperature for 2 hr. The beads were sequentially washed 3× with 0.3% SDS/PBS and then 5× with 25 mM HEPES pH 8.5 (0.5 mL for each wash with 3 min wheel rotation, then centrifuged at 1500 × $g$ for 2 min). Beads were resuspended in 100 μL of 25 mM HEPES pH 8.5.

#### Two-step digestion

Lysates were digested from the beads with 100 ng of LysC (Promega) in 25 mM HEPES pH 8.5 overnight at 37°C with shaking at 1200 RPM in a thermomixer. The supernatant containing eluted peptides was further digested with 100 ng trypsin at 37°C for 6 hr. Samples were acidified with trifluoroacetic acid (TFA) to a final concentration of 0.5% v/v to terminate digestion and frozen at –80°C.

### PerTurboID: Peptide-level sample processing

#### Parasite enrichment

Tightly synchronized PTP4 or KAHRP-TurboID/FIKK4.1 cKO lines were grown in biotin-free media in a 30 mL culture at 3% hematocrit and 5–8% parasitemia, in triplicate using different blood donors. Two cycles before collecting the parasites, the culture was split into two and treated with either 100 nM rapamycin for 4 hr to induce gene knockout, or with DMSO as a control. Then, 100 μM D-biotin (bioAPE) was added two cycles after RAP/DMSO-treatment at 36 hpi for 4 hr. Parasites were Percoll-enriched at 40 hpi, then washed 5× with 5 mL PBS to remove free biotin. Parasites were transferred to 2 mL protein low-bind tubes (Eppendorf, Cat# 022431102) and lysed in 1 mL of 8 M urea in 50 mM HEPES pH 8, with protease (cOmplete mini, Roche) and phosphatase (PhosStop, Roche) inhibitors. Samples were sonicated 3 × 30 s, at a 30% duty cycle on ice, then snap frozen in liquid nitrogen and stored at –80°C.

### Reduction, alkylation, and protein digest

Mass spectrometry-grade reagents and water were used for each step. Parasite lysates were thawed and centrifuged to pellet insoluble material (15 min, 4°C, 21,000 × g). Protein material in the supernatant was quantified with a BCA Protein Assay Kit (Pierce). An equal amount of parasite material was taken from each sample (~2 mg each), and the concentration equalized with 8 M urea in 50 mM HEPES. Oxidized cysteine residues were reduced with 5 mM dithiothreitol (DTT) at room temperature for 60 min. The reduced cysteine residues were then alkylated with 10 mM iodoacetamide in the dark at room temperature for 30 min. For protein digest, samples were diluted 4× with 50 mM HEPES to reduce the concentration of urea to <2 M. Mass spectrometry-grade trypsin was added to a 1:50 enzyme to protein ratio and incubated at 37°C overnight (16 hr). To stop trypsinization, samples were cooled on ice for 10 min, then acidified with TFA at a final concentration of ~0.4% v/v for 10 min on ice. Samples were centrifuged for 15 min at full speed (~21,000 × g) at 4°C to remove insoluble material.

### Sep-Pak desalting

Peptide samples were desalted with Sep-Pak C18 1 cc Vac Cartridges (Waters), one 100 mg column per sample, using a vacuum manifold. Columns were first washed with 3 mL 100% acetonitrile, conditioned with 1 mL 50% acetonitrile and 0.5% acetic acid, and equilibrated with 3 mL 0.1% TFA. Samples were loaded then desalted with 1 mL 0.1% TFA, washed with 1 mL of 0.5% acetic acid, then eluted with 1 mL 50% acetonitrile/0.5% acetic acid. Samples were dried by vacuum centrifugation and stored at –80°C.

### Charging protein G agarose beads with anti-biotin antibodies

60 µL of 50% G agarose bead slurry (Pierce #20398) was used per sample and washed 3× with 5–10 bead volumes of BioSITe buffer (*Kim et al., 2018*) (50 mM Tris, 150 mM NaCl, 0.5% Triton ×100, pH 7.2–7.5 at 4°C). Between washes, beads were centrifuged at 1500 × g, 2 min, 4°C. Anti-biotin antibodies were added (equal mix of Bethyl Laboratories, no. 150-109A, and Abcam no. Ab53494), 30 µg of each antibody per 60 µL of bead slurry, made up to 2 mL with BioSITe buffer, and incubated by rotating at 4°C overnight. Beads were washed 3× with BioSITe buffer, made to an ~50% slurry with BioSITe buffer and aliquoted to 60 µL per sample.

### Biotinylated peptide immunoprecipitation

Dried peptide samples were dissolved in 1.5 mL of BioSITe buffer with vortexing. Samples were neutralized with 1–5 µL of 10 M NaOH to pH 7–7.5 on ice, then centrifuged at 21,000 × g, at 4°C for 10 min to pellet undissolved material. Peptide concentration was quantified with peptide BCA kit (Pierce, # 23275), and an equal amount of each sample was added to the anti-biotin Ab-loaded beads (~60 µL slurry/sample), and incubated rotating for 2 hr at 4°C. Beads were centrifuged (1500 × g, 2 min, 4°C) and washed with 3 × 0.5 mL BioSITe buffer, 1 × 0.5 mL 50 mM Tris, and 3 × 1 mL mass spectrometry-grade water. Biotinylated peptides were eluted from beads with 4 × 50 µL 0.2% TFA and frozen at –80°C.

### Stage tip

Two rounds of Empore C18 membrane were packed into a 200 µL pipette tip and washed with 100 µL methanol by centrifugation (500 × g, 2 min). Tips were equilibrated with 200 µL 1% TFA before loading samples, washing with 200 µL 1% TFA, then eluting with 50 µL of 40% acetonitrile/0.1% TFA. Eluates were dried by vacuum centrifugation and then dissolved in 2% acetonitrile/0.1% formic acid.

### LC-MS/MS (protein-level TurboID and peptide-level PerTurboID)

Samples were loaded onto Evotips (according to the manufacturer's instructions). Following a wash with aqueous acidic buffer (0.1% formic acid in water), samples were loaded onto an Evosep One system coupled to an Orbitrap Fusion Lumos (Thermo Fisher Scientific). The Evosep One was fitted with a 15 cm column (PepSep) and a predefined gradient for a 44 min method was employed. The Orbitrap Lumos was operated in data-dependent mode (1 s cycle time), acquiring IT HCD MS/MS scans in rapid mode after an OT MS1 survey scan (R = 60,000). The MS1 target was 4E5 ions, whereas

the MS2 target was 1E4 ions. The maximum ion injection time utilized for MS2 scans was 300 ms, the HCD normalized collision energy was set at 32, and the dynamic exclusion was set at 15 s.

## Data processing
Acquired raw files were processed with MaxQuant v1.5.2.8 (*Cox and Mann, 2008*).

### TurboID
Peptides were identified from the MS/MS spectra searched against *P. falciparum* (PlasmoDB; *Amos et al., 2022*) and *Homo sapiens* (UniProt; *Bateman et al., 2023*) proteomes using Andromeda (*Cox et al., 2011*) search engine. In addition, an Avidin protein sequence (*Gallus gallus*, UniProt; *Bateman et al., 2023*) was included in the search. Oxidation (M), acetyl (Protein N-term), deamidation (NQ), and acetyl (K) were selected as variable modifications. The enzyme specificity was set to trypsin with a maximum of two missed cleavages.

### PerTurboID
Peptides were identified from the MS/MS spectra searched against *P. falciparum* (PlasmoDB; *Amos et al., 2022*) and *H. sapiens* (UniProt; *Bateman et al., 2023*) proteomes using Andromeda (*Cox et al., 2011*) search engine. Biotin (K), oxidation (M), acetyl (Protein N-term), and phospho (STY) were selected as variable modifications, whereas carbamidomethyl (C) was selected as a fixed modification. The enzyme specificity was set to trypsin with a maximum of three missed cleavages. Minimal peptide length was set at six amino acids. Biotinylated peptide search in Max Quant was enabled by defining a biotin adduct (+226.0776) on lysine residues as well as its three diagnostic ions: fragmented biotin (m/z 227.0849), immonium ion harboring biotin with a loss of $NH_3$ (m/z 310.1584), and an immonium ion harboring biotin (m/z 327.1849).

### TurboID and PerTurboID
The precursor mass tolerance was set to 20 ppm for the first search (used for mass re-calibration) and to 4.5 ppm for the main search. The datasets were filtered on posterior error probability (PEP) to achieve a 1% false discovery rate on protein, peptide, and site level. Other parameters were used as preset in the software. 'Unique and razor peptides' mode was selected to allow identification and quantification of proteins in groups (razor peptides are uniquely assigned to protein groups and not to individual proteins). Intensity-based absolute quantification (iBAQ) in MaxQuant was performed using a built-in quantification algorithm (*Cox and Mann, 2008*) enabling the 'Match between runs' option (time window 0.7 min) within replicates. Label-free quantitation (LFQ) intensities were also acquired for FIKK4.1 TurboID but these were not used in our analysis.

## Data analysis
MaxQuant output files were processed with Perseus, v1.5.0.9 (*Tyanova et al., 2016*), Microsoft Office Excel 2016, and using R programming in R studio. The R packages used were readxl, xlsx, matrixStats, gridExtra, ggplot2, gplots, stringr, plyr, data.table, and svglite.

### TurboID
For the FIKK4.1-TurboID protein-level dataset, protein group data were filtered to remove contaminants, protein IDs originating from reverse decoy sequences, and those only identified by site. iBAQ intensities were log2 transformed, and peptides with one or less valid value were removed. The iBAQ intensities were normalized by median ratio normalization (*Love et al., 2014*). Briefly, the average log2 intensities across conditions were calculated for each unique biotinylated peptide (row means). This row mean was subtracted from each individual log2 intensity in a row, and a median was taken of each column of this new dataset (scaling factor). The scaling factor was then subtracted from all values in the corresponding column of the original dataset to give normalized log2 intensities.

Only proteins which were more abundant in the FIKK4.1-TurboID datasets relative to the NF54 control were included as other proteins are likely to represent background binding to the beads. Inclusion was based on the following criteria: either the average intensity of all FIKK4.1-TurboID samples > average of NF54 samples +1.5, or all intensity values were missing from NF54 samples, and all three values present in either the FIKK4.1 late or FIKK4.1 early replicate samples. Missing values were

imputed where necessary to allow for the calculation of fold changes. If two of three replicates had a detectable signal in a given condition, the missing value was imputed based on the average of the other two values. Intensity values still missing after this imputation step were imputed from a shifted normal distribution based on the method used in Perseus software (*Tyanova et al., 2016*), with a mean intensity downshifted by 1.8, and the standard deviation shrunken to 30% of the original.

The log2 fold change between FIKK4.1-turboID early and late samples and NF54 samples were calculated and p-values were acquired by two-sample *t*-tests assuming equal variances, n = 3. The p-values were corrected for multiple comparisons using the Benjamini–Hochberg method, with a false discovery rate significance threshold of <10% (-log10(q-value) > 1).Only proteins where the log2 fold change TurboID/NF54 for either early or late timepoints >1.5, and where the average of all TurboID/NF54 >1.5, were included. Log2 fold change values for early vs. late samples were then calculated.

### PerTurboID

For the PerTurboID peptide-level dataset, modified peptide data were filtered to remove contaminants and IDs originating from reverse decoy sequences. Non-biotinylated peptides (background) were then removed from the dataset. The resulting biotinylated peptide list was annotated with biotin (K) and phospho (STY) site information. IBAQ intensities were normalized by median ratio normalization as described above, with KAHRP and PTP4-turboID datasets normalized separately. If two of three replicates in a given condition had a detectable signal, the third missing value was imputed based on the average of the other two values. In cases where two conditions were compared (e.g. KAHRP RAP vs. KAHRP DMSO), missing values were imputed from a shifted normal distribution if the following criteria applied: sum(condition1)/3 > sum(condition2) + 1, and sum(condition1) > 50. The normal distribution was calculated from all intensities for a given peptide based on the method used in Perseus software as above (*Tyanova et al., 2016*). This allowed peptide intensities to be compared where there were three high valid values in one condition and one low value or no values in another as expected where large differences are observed between conditions.

The log2 fold change between DMSO and RAP-treated KAHRP or PTP4-turboID lines was calculated only where no missing values were present in either condition after imputation, and p-values were acquired by two-sample *t*-tests assuming equal variances, n = 3. Several peptides often represented the same biotinylation site due to the presence of multiply biotinylated peptides, missed protease cleavage, and post-translational modifications. For directly comparing the extent of biotinylation on a specific site, the log2 fold change values for all peptides representing the same site were averaged, and p-values were combined according to Fisher's method. Combined p-values were corrected for multiple comparisons using the Benjamini–Hochberg method. These corrected q-values were -log10 transformed for all plots, with a false discovery rate significance threshold of <10% (-log10(q-value) > 1).

For comparing protein intensities in the KAHRP vs. PTP4 dataset, the same data analysis pipeline as above was used, except that KAHRP DMSO and PTP4 DMSO intensities were compared rather than RAP vs. DMSO. Statistics were performed in the same way.

The mass spectrometry proteomics data have been deposited to the ProteomeXchange Consortium via the PRIDE (*Perez-Riverol et al., 2019*) partner repository with the dataset identifier PXD039125.

### Network analysis

An undirected network was constructed in Cytoscape (*Shannon et al., 2003*) with either KAHRP, PTP4, or FIKK4.1 as main nodes connected to other nodes that represent proteins observed in their respective TurboID datasets. Only proteins with intensities over the background NF54 line were included for FIKK4.1, according to the criteria described in the 'Data analysis section.

### RHOPH complex structural analysis

The structure of the RHOPH complex was visualized with Chimera software (*Pettersen et al., 2004*). The cryo-electron microscopy structure (PDB: 7KIY; *Schureck et al., 2021*) was superimposed with AlphaFold-predicted structures to represent unresolved regions within the protein structure (*Jumper et al., 2021*).

## Flow cytometry analysis of Var2CSA and RIFIN surface expression

For flow cytometry analysis of Var2CSA surface expression, mixed culture parasites at ~5–10% parasitemia were first blocked with 1% BSA in PBS for 30 min, then incubated with either rabbit anti-Var2CSA antibodies (*Srivastava et al., 2011*) (a kind gift from Benoit Gamain, INSERM Paris) or human anti-Var2CSA antibodies (*Barfod et al., 2007*) (a kind gift from Lars Hviid, University of Copenhagen) at a dilution of 1:100 for 1 hr in 1% BSA at room temperature (*Barfod et al., 2007*). Cells were washed 3× with 1% BSA before incubating for 1 hr in anti-human IgG secondary antibody labeled with either APC (BioLegend, #410712, dilution 1:100) or Phycoerythrin (Abcam, #ab98596, dilution 1:100) in 1% BSA, in the dark. iRBC were washed 3× with PBS then fixed in 4% paraformaldehyde, 0.2% glutaraldehyde in PBS, for 30 min at room temperature. Samples were then washed once with PBS and resuspended in PBS with Hoechst 33342 dye (New England Biolabs) (1:1000 dilution) and incubated at 37°C for 20 min. Parasites were run on a BD LSR Fortessa cell analyzer. Phycoerythrin fluorescence was detected with a 561 nm (yellow) excitation laser and a 586/15 bandpass filter. APC fluorescence was detected with a 640 nm (red) excitation laser and a 670/14 bandpass filter. Hoechst fluorescence was detected with a 355 nm (UV) excitation laser and a 450/50 bandpass filter. Flow cytometry data was analyzed with FCS express software, gating first on single RBCs, then infected RBCs (Hoechst positive), then PE or APC-positive iRBC. The median fluorescence intensity was calculated for this population. For comparing Var2CSA surface expression between FIKK WT and KO lines, parasites were treated with rapamycin or DMSO 72 hr before conducting the flow cytometry experiments, and the ratio of median fluorescence intensity of Var2CSA for RAP/DMSO was calculated.

For flow cytometry analysis of surface-exposed RIFINs, recombinant LAIR1-Fc or LILRB1-Fc proteins (*Saito et al., 2017*) (a kind gift from Hisashi Arase, Osaka University) were pre-incubated with anti-human-Fc secondary antibodies fused to APC (BioLegend, #410712, dilution 1:100) for 1 hr in 1% BSA in PBS. This mixture was then incubated with iRBC for 1 hr at room temperature, before washing 3× with PBS, fixing in 4% paraformaldehyde, 0.2% glutaraldehyde in PBS for 30 min, and running on the flow cytometer as described for analyzing Var2CSA surface expression, above.

## HBEC-5i and CD36-CHO cell culture

HBEC-5i cells (a kind gift from Maria Bernabeu, EMBL Spain) were cultured in 0.1% gelatin-coated (Sigma-Aldrich) culture flasks with DMEM-F12 (Gibco) supplemented with 20 μg/mL endothelial cell growth supplement (Gibco) and 10% fetal bovine serum (FBS, Gibco). Chinese Hamster Ovary (CHO) cells expressing CD36 (*Lubiana et al., 2020*) (a kind gift from Iris Bruchhaus, Bernhard Nocht Institute for Tropical Medicine) were cultured in Hams F12 media (Gibco) with 10% FBS.

## Cytoadhesion assay

EPCR/ICAM1-binding iRBCs were obtained via magnetic enrichment of dual-binding iRBCs with ProteinA-coupled Dynabeads (Invitrogen), which were associated to anti-EPCR/ICAM1-binding PfEMP1 antibodies (a kind gift from Anja Jensen, University of Copenhagen) as described previously (*Joergensen et al., 2010*). Late trophozoites were selected for CD36 binding by incubating Percoll-enriched iRBCs with CD36-CHO cells for 1 hr at 37°C as previously described (*Lubiana et al., 2020*). Adherent cells were kept in culture after 5× washing with RPMI. At least three rounds of selection were performed for enrichment of either CD36- or HBEC-5i-binding parasites. HBEC-5i and CD36-CHO cells were seeded on ibiTreat μ-slide VI$^{0.4}$ (ibidi) 2 d prior to the assay for confluency. Percoll-enriched DMSO or rapamycin-treated iRBCs (36–40 hpi) were DNA-stained with either Hoechst 33342 (New England Biolabs) or SYBRGreen and mixed in a 1:1 ratio in duplicate. IRBCs were then flowed over the HBEC-5i or CD36-CHO seeded cells in the microchannel for 15 min at a flow rate of 0.13 dyn/cm$^2$ (4.5 mL/hr). Ten frames across the microchannel were taken with Nikon Microscope. Cytoadherent cells were quantified via ImageJ imaging software. The adhesion frequency was calculated by normalizing the rapamycin-treated samples to the ratio of the DMSO baseline control. Experimental data were analyzed using GraphPad Prism 8 software.

For measuring adhesion to CSA, FIKK4.1 cKO parasite lines naturally expressing Var2CSA were used, as confirmed by flow cytometry. IbiTreat μ-slide VI$^{0.4}$ slides (ibidi) were coated with 10 mg/mL CSA for 2 hr then washed with 50 mL PBS. Percoll-enriched DMSO or rapamycin-treated synchronized iRBCs were stained with either Hoechst 33342 (New England Biolabs) or SYBRGreen and mixed in a 1:1 ratio in duplicate, as above. IRBCs were flowed over the slides for 10 min at a flow rate

of 0.13 dyn/cm² (4.5 mL/hr), before gradually increasing the flow rate up to 4.4 dyn/cm² (150 mL/hr). Fifteen frames across the microchannel were taken with a Nikon Microscope at each flow rate. Quantification of cytoadherent cells was automated with ImageJ macros to first filter for cells then quantify the blue/green fluorescence intensity in each cell to identify the differentially stained RAP or DMSO-treated parasites. The adhesion frequency was calculated by calculating the ratio of RAP/DMSO-treated parasites. The initial RAP/DMSO ratio was calculated by applying parasites directly to the slides with no flow and analyzed the same way. Experimental data were analyzed using GraphPad Prism 8 software.

## Acknowledgements

We thank all members of the Treeck lab for critical input. We would like to thank Benoit Gamain, Lars Hviid, and Anja Remsted-Jensen for providing antibodies against PfEMP1 variants. We thank Hisashi Arase for reagents to detect LILRB1 and LAIR1-binding RIFINs, Maria Bernabeu for HBEC-5i cells, and Iris Bruchhaus for Chinese Hamster Ovary (CHO) cells expressing CD36. We would like to thank the Crick Science Technology platforms (STPs) (Proteomics, Flow Cytometry, Biological Safety, and Light Microscopy) for support. Special thanks to PlasmoDB providing a critical resource (*Amos et al., 2022*). MT received funding from the Francis Crick Institute which receives its core funding from Cancer Research UK (CC2132), the UK Medical Research Council (CC2132), and the Wellcome Trust (CC2132). The ScienceTechnology Proteomics Platform at the Francis Crick Institute received funding from Cancer Research UK (CC0199), The UK Medical Research Council (CC0199) and the Wellcome Trust (CC0199). For the purpose of Open Access, the authors have applied a CC BY public copyright license to any Author Accepted Manuscript version arising from this submission.

## Additional information

### Funding

| Funder | Grant reference number | Author |
| --- | --- | --- |
| Francis Crick Institute | CC2132 | Heledd Davies<br>Hugo Belda<br>Malgorzata Broncel<br>Jill Dalimot<br>Moritz Treeck |
| Francis Crick Institute | CC0199 | Malgorzata Broncel<br>Moritz Treeck |

The funders had no role in study design, data collection and interpretation, or the decision to submit the work for publication.

### Author contributions

Heledd Davies, Conceptualization, Software, Formal analysis, Investigation, Visualization, Methodology, Writing – original draft, Writing – review and editing; Hugo Belda, Malgorzata Broncel, Formal analysis, Investigation, Methodology, Writing – review and editing; Jill Dalimot, Formal analysis, Investigation, Writing – review and editing; Moritz Treeck, Conceptualization, Formal analysis, Supervision, Funding acquisition, Writing – original draft, Project administration, Writing – review and editing

### Author ORCIDs

Heledd Davies (ID) http://orcid.org/0000-0003-3475-0080
Malgorzata Broncel (ID) http://orcid.org/0000-0003-2991-3500
Jill Dalimot (ID) http://orcid.org/0000-0002-1306-093X
Moritz Treeck (ID) http://orcid.org/0000-0002-9727-6657

### Decision letter and Author response

Decision letter https://doi.org/10.7554/eLife.86367.sa1
Author response https://doi.org/10.7554/eLife.86367.sa2

## Additional files

### Supplementary files
• Supplementary file 1. Unfiltered raw data for FIKK4.1-TurboID experiment.

• Supplementary file 2. Processed FIKK4.1 TurboID data.

• Supplementary file 3. Unfiltered and unprocessed data for KAHRP and PTP4 PerTurboID experiment.

• Supplementary file 4. KAHRP and PTP4 PerTurboID processed site-level data.

• Supplementary file 5. KAHRP and PTP4 PerTurboID processed peptide-level data.

• Supplementary file 6. Network analysis of proteins observed in FIKK4.1, KAHRP, and PTP4 TurboID.

• Supplementary file 7. PHIST proteins in dataset.

• Supplementary file 8. KAHRP vs. PTP4 TurboID peptide-level data.

• Supplementary file 9. KAHRP and PTP4-TurboID phosphorylated proteins.

• Supplementary file 10. Primer list.

• MDAR checklist

### Data availability
The mass spectrometry proteomics data have been deposited to the ProteomeXchange Consortium via the PRIDE partner repository with the dataset identifier PXD039125. Source data for all Western Blots and PCRs is shown in two separate Source data files.

The following dataset was generated:

| Author(s) | Year | Dataset title | Dataset URL | Database and Identifier |
|---|---|---|---|---|
| Treeck M | 2022 | PerTurboID of *Plasmodium falciparum* cytoadhesion complex | https://www.ebi.ac.uk/pride/archive/projects/PXD039125 | PRIDE, PXD039125 |

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
