## [Editor Report]

This important study combines conditional mutagenesis with proximity labeling to evaluate alterations in a sub-cellular proteome upon a perturbing event. The approach is applied to the deletion of a kinase involved in trafficking of adhesins to the malaria parasite-infected erythrocyte surface and the evidence supporting the conclusions is compelling. The work will be of broad interest to cell biologists and biochemists.

---

## [Decision Letter]

**Decision letter after peer review:**

Thank you for submitting your article "PerTurboID: A targeted in situ method to measure changes in a local protein environment reveals the impact of kinase deletion on cytoadhesion in malaria causing parasites" for consideration by *eLife*. Your article has been reviewed by 3 peer reviewers, and the evaluation has been overseen by a Reviewing Editor and Dominique Soldati-Favre as the Senior Editor. The reviewers have opted to remain anonymous.

Essential revisions:

The study is sophisticated and well-carried out. Despite some the limitations of the methods that prevent conclusive statements on the precise mechanism of PfEMP1 trafficking and membrane insertion, the findings are considered valuable for the community.

The reviewers have not raised any major issues but several specific points that will need your attention. They have clearly expressed their concerns so I will not reiterate them here.

However, and importantly, the presentation is confusing and could be improved. Some of the figures are to be revised to improve clarify and the discussion should address the caveats and experimental limitations.

*Reviewer #1 (Recommendations for the authors):*

1. Line 174. Description of the TurboID tagging of PTP4 or KAHRP in the FIKK4.1 conditional KO line. It was initially unclear to this reviewer that the conditional KO was made first, cloned and then used for TurboID tagging. This should be spelt out for readers who may otherwise be confused.

2. Supp. Figure 1

– Constructs should be listed in the order they are presented in the paper. ie. FIKK4.1 TurboID first.

– Coloring of primers to indicate position, while concise, may be difficult for some readers, esp. considering that 8% of males are red-green color-blind. I suggest using primer numbers.

3. Supp. Figure 2B – anti-HA blots for FIKK4.1 show many bands, when there should be one at ~74 kDa for FIKK4.1. The second most prominent band is below 100 kDa could reflect addition of a 30 kDa addition of NeoR for the selection-linked integration. The authors should clarify this. In this reviewer's hands, T2A cleavage is generally efficient. However, could it be compromised in the case of this exported protein?

4. Figure 4A. Barcoding of the results from + and – rapamycin treatment to disrupt FIKK4.1 is an elegant way to show these complex data. However, the presentation is confusing and could be improved. For KAHRP, I understand that FIKK4.1 KO will reduce phosphorylation in the 351-364 region and that therefore unphosphorylated peptides that include this region will become more abundant. However, it is not clear that the fraction of the unphosphorylated peptides that get biotinylated will necessarily increase. For each row in the case of KAHRP, is the fold change representation calculated relative to the corresponding pool (i.e. the biotinylated fraction of the unphosphorylated pool for the minus P column and the biotinylated fraction of the phosphorylated pool for the plus P column)? Alternatively, does it represent fold change relative to the total amount of the peptide (phos + unphos together)? This second calculation is misleading, I think. Why was it necessary to split KAHRP into minus and plus P arms, but not the other proteins displayed here? Presumably, for each lysine, there is an unphosphorylated fraction even when FIKK4.1 is not disrupted because of finite kinase efficiency.

Another issue with the barcodes: For ACS7 and RhoPH3 positions 876 and 853 are mentioned in the Results. However, numbered positions are not marked on the column plots, so the reader is left wondering which red or blue line in each column is being described. It might help to label critical positions with numbers on each column.

Would it help to have two sets of bar codes for each protein, one showing the fold change in phosphorylation upon FIKK4.1 knockout and the other showing the fold change in biotinylation (calculated separately for phosphorylated and unphosphorylated pools)?

5. Line 522-527. The authors only briefly mention caveats with use of PerTurboID and then state the limitations are "largely overcome by the alternative processing method". Given the limitations, which are described above in my review, I would suggest expanding the discussion of the caveats and use phraseology more cautiously than "largely overcome". Otherwise, the Discussion section is appropriately cautious.

*Reviewer #2 (Recommendations for the authors):*

– Line 27: a citation to the current world malaria report should be included here since these numbers change with years.

– Figure 1B, 2D-G, S3: It would help the reader to interpret these images if the authors could include a transmitted light channel to orient to the RBC surface vs parasite periphery/PV.

– Figure S1: The FIKK4.1 turboID tagging schematic shows a stop codon is retained at the end of the FIKK4.1 sequence after modification of the locus to produce the tagged version of the gene. I assume this is an error and that only the stop codon after V5 should be present. Also, there should be a stop codon after the NeoR in the FIKK4.1 kinase deletion mutant (assuming this is the same strain/design as in the authors' previous Nat Micro paper) here and in the schematic in Figure 2A. Finally, it might help the reader to have the schematics in Figure S1 presented in the order they are discussed in the main text (ie, FIKK4.1-TurboID first, etc).

– Line 126: Needs a citation for this claim that ~16% of the genome encodes predicted exported proteins.

– Line 129-131: The authors may wish to note that RBCs are known to be the major peripheral source of α-synuclein (Barbour et al., 2008; PMID: 18182779). Although to my knowledge a function for α synuclein in RBCs is not known, this may help to explain why it was detected in these experiments. Given α synuclein is highly enriched in the KAHRP-T dataset, does this suggest localization at the RBC membrane/cytoskeleton? If so, are there any implications for RBC remodeling? Since the α synuclein data are mentioned multiple times in the Results section, the authors might comment briefly on this in the discussion.

– Line 153-156, Figure 1G: Human proteins are better represented at the 16-20 hpi time point while exported parasite proteins are better represented at the 40-44 hpi time point and non-exported parasite proteins seem generally evenly distributed between early and late time points. Does RBC biotin permeability change during intraerythrocytic development (perhaps based on activation of NPPs beginning at ~14 hpi and increasing into the trophozoite stage)? If so, how might higher biotin permeation at the later time point influence the data?

– Lines 216, 269, 277, 279, 494 496, 498, 807: This complex is generally referred to as the "RhopH complex" rather than the PSAC complex, since although its components are important to PSAC channel function, the exact mechanism has not been established (ie, whether RhopH directly forms the channel, etc).

– Line 274: Although not used to map their site-specific proteomic data, the authors might also wish to cite the independent soluble RhopH complex structure by Ho et al. 2021 (PMID: 34446549) here.

– Line 289: The authors should support this claim by citing Guy et al. 2015, PMID: 26513658.

– Line 299: Capitalize "turboID" as with other occurrences throughout the manuscript.

– Figure 1C, 2B: The data in 1C show that surface presentation of Var2CSA is not diminished by Turbo activity when biotin is added to FIKK4.1-T parasites. Was this assay ever carried out on the unmodified WT parental parasites to ensure the Turbo fusion itself to FIKK4.1 does not impact Var2CSA presentation? Similarly, Figure 2B shows that the TurboID tag on PTP4 or KAHRP does not change surface presentation relative to the FIKK4.1 diCre kinase domain mutant background on which these lines were built. However, given this FIKK4.1 conditional mutagenesis strategy involves at T2A skip after the HA epitope tag which does not skip with 100% efficiency (meaning that a portion of the FIKK4.1 protein will contain a C-terminal fusion to NeoR that could alter function, as shown in the western blots from the authors' previous Nat Micro paper and apparent here in the FIKK4.1 band detected by anti-HA at a higher than expected MW in Figure S2B which appears to account for ~30% of the FIKK4.1 protein) it seems important to compare with Var2CSA surface exposure in the parental (WT unmodified) line.

– Figure 1F: This is a nice diagram to convey the organization of the RBC membrane/cytoskeleton and summarize the top human proteins identified by FIKK4.1-T. However, I would recommend color coding the parasite proteins as well – or at least using a different color than the grey that is used for the human proteins that were not enriched by PL since this might confusingly suggest that PfEMP1 and other parasite-encoded knob proteins were not enriched by FIKK4.1-T.

– Figure 2B: Please clarify if these Var2CSA surface presentation assays were carried out in the presence or absence of biotin.

– Figure S2A: several major biotinylated bands in the KAHRP-TurboID input are noticeably absent in the avidin pull down; most notable, two major bands at about 80 and 90 kD and a band >135 kD. Curiously, these bands also appear much less prominent in the time course in S2C. Any comment on why this might be?

– Line 464: this should be "KO/WT" as in the Y-axis of 5F.

– Lines 507-514: It is also possible that other FIKKs might function redundantly to support PfEMP1 trafficking.

In Figure 3C, the "RBC cytosol" label should be added to the bottom of the PV group as in the "RBC surface" and "Maurer's Clefts" groups to avoid possible confusion about topology. Also, J dots (which should probably be labeled "J dots" instead of just "Dots" in keeping with conventional nomenclature), are thought to be membrane-less organelles and thus the membrane in the diagram may be confusing. Additionally, the authors should clarify where topology has been biochemically determined (I only found this for SBP1 with refs 54 and 55, lines 520-521) vs where it is suggested by their biotinylated peptide data.

– Figure 5A: It may be useful to the reader to note in the figure legend that the FIKK9s mutant deletes 7 FIKKS while all others are individual FIKK knockouts.

– Line 926: The DMSO condition is not technically "WT" as it contains a 3xHA-2A-NeoR tag on FIKK4.2.

*Reviewer #3 (Recommendations for the authors):*

The paper is well written, and the data is very clear. There are a few suggestions to improve the work further:

1. Some of the figures are very busy and contain many panels. This makes it difficult to sometimes follow.

2. For IFA images shown it would be good to also show light microscopy to clearly show the whole cell as well.

3. As far as I understand disruption of FIKK4.1 leads to var2CSA not being translocated to the surface of the iRBC. It would therefore not assemble into knobs. Also, it means that it would accumulate extensively in the MCs. This raises a number of questions in relation to the interpretation of the data:

a. Are changes observed a result of a true overall change in structure or conformation or rather represent the fact that there is now a gap (due to missing var2CSA) that now creates accessibility.

b. Does the absence of var2CSA on the surface change overall RBC membrane characteristics that now for example allow PSAC to become more mobile and therefore come into closer proximity to the tagged proteins?

c. Are changes observed in the biotinylation of var2CSA a result of it getting stuck in MCs and if this is the case wouldn't one expect to see this also in the presence of FIKK4.1 as much of PfEMP1 remains in MCs.

4. In Figure 5. The authors show that FIKK4.1 impacts surface location of PfEMP1 but not RIFIN. To achieve this the authors used tagged LILRB1 and LAIR1 to demonstrate this. However, there are a number of issues that need to be considered.

a. In line 434 the authors state that RIFINs are surface exposed and cite the Saito et al., 2017 paper in support. However, as far as I understand the authors do not directly demonstrate that RIFINs are on the surface of the iRBC. They only show binding of the LILRB1 to the surface of the infected RBC. Moreover, the authors show that LILRB1 binding only happens in a small fraction of iRBC and is rapidly lost upon culturing. Can the authors be certain that the signal they detect is indeed due to RIFIN?

b. Since there is only a small subset of RIFIN that bind either LAIR or LILRB1 it would be interesting to note their expression level or whether disruption of FIKK4.1 has any impact on them.

[Editors' note: further revisions were suggested prior to acceptance, as described below.]

Thank you for resubmitting your work entitled "PerTurboID, A targeted in situ method reveals the impact of kinase deletion on its local protein environment in the cytoadhesion complex of malaria causing parasites" for further consideration by *eLife*. Your revised article has been evaluated by Dominique Soldati-Favre (Senior Editor) and a Reviewing Editor.

The manuscript has been improved and will be acceptable for publication upon a minor revision addressing some remaining issues as outlined below:

*Reviewer #3 (Recommendations for the authors):*

Overall, the authors have addressed the issues raised by the reviewers.

However, I am still not completely convinced by the arguments of the authors in relation to RIFIN surface expression.

I have looked at the presented data carefully again and also had another look at the Harrison et al. paper mentioned.

While there is little doubt that some RIFIN interact with LLIRB and LAIR there is no demonstration in any of the papers I am aware of that these RIFIN are actually expressed on the surface of the iRBC. NK cell inactivation can be achieved by secretion via microvessels or other mechanisms that do not involve surface expression. Moreover, the MFI is just around 100 as compared to around 600 for var2CSA expression. This raises some questions on how specific to RIFIN the observed signal really is.

I would recommend that the authors take this into consideration and tune down the conclusion a bit.

---

## [Author Response]

Essential revisions:Reviewer #1 (Recommendations for the authors):1. Line 174. Description of the TurboID tagging of PTP4 or KAHRP in the FIKK4.1 conditional KO line. It was initially unclear to this reviewer that the conditional KO was made first, cloned and then used for TurboID tagging. This should be spelt out for readers who may otherwise be confused.

This has been clarified.

2. Supp. Figure 1– Constructs should be listed in the order they are presented in the paper. ie. FIKK4.1 TurboID first.– Coloring of primers to indicate position, while concise, may be difficult for some readers, esp. considering that 8% of males are red-green color-blind. I suggest using primer numbers.

Constructs have been rearranged, and primer numbers have been added.

3. Supp. Figure 2B – anti-HA blots for FIKK4.1 show many bands, when there should be one at ~74 kDa for FIKK4.1. The second most prominent band is below 100 kDa could reflect addition of a 30 kDa addition of NeoR for the selection-linked integration. The authors should clarify this. In this reviewer's hands, T2A cleavage is generally efficient. However, could it be compromised in the case of this exported protein?

Yes, we have observed non-cleaved T2A cleavage for almost all FIKKs. The nature of this remains unknown, but could be related to export. We have added a comment to the figure legend.

4. Figure 4A. Barcoding of the results from + and – rapamycin treatment to disrupt FIKK4.1 is an elegant way to show these complex data. However, the presentation is confusing and could be improved. For KAHRP, I understand that FIKK4.1 KO will reduce phosphorylation in the 351-364 region and that therefore unphosphorylated peptides that include this region will become more abundant. However, it is not clear that the fraction of the unphosphorylated peptides that get biotinylated will necessarily increase. For each row in the case of KAHRP, is the fold change representation calculated relative to the corresponding pool (i.e. the biotinylated fraction of the unphosphorylated pool for the minus P column and the biotinylated fraction of the phosphorylated pool for the plus P column)? Alternatively, does it represent fold change relative to the total amount of the peptide (phos + unphos together)? This second calculation is misleading, I think. Why was it necessary to split KAHRP into minus and plus P arms, but not the other proteins displayed here? Presumably, for each lysine, there is an unphosphorylated fraction even when FIKK4.1 is not disrupted because of finite kinase efficiency.

We have made a schematic to illustrate the effect of changes in phosphorylation on the log2 fold change RAP/DMSO ratio. This demonstrates that it is not necessary for biotinylation to change for us to observe a change in the ratio. In fact, we assume that the accessibility to biotinylation does not change, but that the pool of unphosphorylated protein increases. As we don’t observe the non-biotinylated peptides at all, all the fold changes are simply RAP/DMSO, calculated separately for the phosphorylated and non-phosphorylated version of the peptide (not relative to the total amount). Only the phosphorylated peptides for KAHRP are shown in the main figure in order not to overcomplicate it, as KAHRP best illustrates the points discussed. All other proteins with at least one significantly changing peptide are shown in Supplementary figure 5, where all proteins with observed phosphorylated peptides include a +P barcode. For the main figure, the phosphorylated peptides for PTP4, ACS7, RHOPH3, and PF3D7_0702500 are not shown, rather than combined with the non-phosphorylated peptide ratios.

Another issue with the barcodes: For ACS7 and RhoPH3 positions 876 and 853 are mentioned in the Results. However, numbered positions are not marked on the column plots, so the reader is left wondering which red or blue line in each column is being described. It might help to label critical positions with numbers on each column.

We have included asterisks in the figure to indicate all positions discussed in the main text. The tables provide the position of all observed peptides.

Would it help to have two sets of bar codes for each protein, one showing the fold change in phosphorylation upon FIKK4.1 knockout and the other showing the fold change in biotinylation (calculated separately for phosphorylated and unphosphorylated pools)?

We have added a table of all peptides for which phosphorylated peptides are detected, with a ratio non-phosphorylated/phosphorylated (table S9). The caveat for this is that it cannot truly represent the ratio of phosphorylation as it assumes no change in biotinylation, which is unlikely to be the case.

5. Line 522-527. The authors only briefly mention caveats with use of PerTurboID and then state the limitations are "largely overcome by the alternative processing method". Given the limitations, which are described above in my review, I would suggest expanding the discussion of the caveats and use phraseology more cautiously than "largely overcome". Otherwise, the Discussion section is appropriately cautious.

The words ‘largely overcome’ applied to the limitations of traditional turboID in assigning a threshold of ‘biotinylation’ based on peptide intensity relative to the no-biotin/wildtype background sample, as sticky peptides are enriched by the beads. This no longer applies to PerTurboID as biotinylation is confirmed by the mass spectrometer. We have expanded the discussion to address some limitations discussed by the reviewer above.

Reviewer #2 (Recommendations for the authors):– Line 27: a citation to the current world malaria report should be included here since these numbers change with years.

This has been added.

– Figure 1B, 2D-G, S3: It would help the reader to interpret these images if the authors could include a transmitted light channel to orient to the RBC surface vs parasite periphery/PV.

A new supplementary image has been created with all brightfield images. We excluded them in the first submission as MeOH fixation was used which does not retain the morphology of the cell well. In addition, the iSIM microscope which was used, while increasing resolution for the immune-fluorescence images, provides reduced quality for the brightfield. We now include the brightfield images in the supplementary figure to show an approximation of the RBC boundary for the main images, represented by a dotted line.

– Figure S1: The FIKK4.1 turboID tagging schematic shows a stop codon is retained at the end of the FIKK4.1 sequence after modification of the locus to produce the tagged version of the gene. I assume this is an error and that only the stop codon after V5 should be present. Also, there should be a stop codon after the NeoR in the FIKK4.1 kinase deletion mutant (assuming this is the same strain/design as in the authors' previous Nat Micro paper) here and in the schematic in Figure 2A. Finally, it might help the reader to have the schematics in Figure S1 presented in the order they are discussed in the main text (ie, FIKK4.1-TurboID first, etc).

Yes this was a mistake, and has been corrected. The order has also been changed.

– Line 126: Needs a citation for this claim that ~16% of the genome encodes predicted exported proteins.

This number was incorrect, and has been changed to 8% with a citation added.

– Line 129-131: The authors may wish to note that RBCs are known to be the major peripheral source of α-synuclein (Barbour et al., 2008; PMID: 18182779). Although to my knowledge a function for α synuclein in RBCs is not known, this may help to explain why it was detected in these experiments. Given α synuclein is highly enriched in the KAHRP-T dataset, does this suggest localization at the RBC membrane/cytoskeleton? If so, are there any implications for RBC remodeling? Since the α synuclein data are mentioned multiple times in the Results section, the authors might comment briefly on this in the discussion.

We agree that is it likely that α-synuclein localises to the RBC cytoskeleton, and have added this and the citation to the discussion.

– Line 153-156, Figure 1G: Human proteins are better represented at the 16-20 hpi time point while exported parasite proteins are better represented at the 40-44 hpi time point and non-exported parasite proteins seem generally evenly distributed between early and late time points. Does RBC biotin permeability change during intraerythrocytic development (perhaps based on activation of NPPs beginning at ~14 hpi and increasing into the trophozoite stage)? If so, how might higher biotin permeation at the later time point influence the data?

This is an interesting possibility and while it’s possible that NPP activity affects biotin permeability, we think it’s unlikely due to the small size of biotin and as we observe a biotin signal in Turbo-ID lines by IFA before ~14 hpi. Regardless, as the TurboID tag resides within the cell, a change in biotin permeability would affect all proteins equally and wouldn’t affect the data as far as we are aware.

– Lines 216, 269, 277, 279, 494 496, 498, 807: This complex is generally referred to as the "RhopH complex" rather than the PSAC complex, since although its components are important to PSAC channel function, the exact mechanism has not been established (ie, whether RhopH directly forms the channel, etc).

Thank you we have changed PSAC to RHOPH complex.

– Line 274: Although not used to map their site-specific proteomic data, the authors might also wish to cite the independent soluble RhopH complex structure by Ho et al. 2021 (PMID: 34446549) here.

Yes both structures should certainly be cited, thank you for bringing this oversight to our attention.

– Line 289: The authors should support this claim by citing Guy et al. 2015, PMID: 26513658.

Citation added.

– Line 299: Capitalize "turboID" as with other occurrences throughout the manuscript.

Done.

– Figure 1C, 2B: The data in 1C show that surface presentation of Var2CSA is not diminished by Turbo activity when biotin is added to FIKK4.1-T parasites. Was this assay ever carried out on the unmodified WT parental parasites to ensure the Turbo fusion itself to FIKK4.1 does not impact Var2CSA presentation? Similarly, Figure 2B shows that the TurboID tag on PTP4 or KAHRP does not change surface presentation relative to the FIKK4.1 diCre kinase domain mutant background on which these lines were built. However, given this FIKK4.1 conditional mutagenesis strategy involves at T2A skip after the HA epitope tag which does not skip with 100% efficiency (meaning that a portion of the FIKK4.1 protein will contain a C-terminal fusion to NeoR that could alter function, as shown in the western blots from the authors' previous Nat Micro paper and apparent here in the FIKK4.1 band detected by anti-HA at a higher than expected MW in Figure S2B which appears to account for ~30% of the FIKK4.1 protein) it seems important to compare with Var2CSA surface exposure in the parental (WT unmodified) line.

We have included supplementary figures comparing FIKK4.1 cKO Var2CSA presentation to that of wildtype, demonstrating that there is very little difference. Because this FIKK4.1 cKO line is the parent of the KAHRP and PTP4-TurboID lines it is used as a control for all other experiments.

We did not include a wild-type control for our FIKK4.1-TurboID flow cytometry experiments as the wild-type unfortunately switched PfEMP1 variants away from Var2CSA. In lieu of that, we included several other parasite lines including FIKK4.2 cKO, FIKK7.1 cKO, and FIKK4.1 cKO, which are all wildtype at the FIKK4.1 locus. While the FIKK4.1-TurboID had lower median fluorescence intensity for Var2CSA compared to FIKK4.2 and FIKK4.1 cKO lines, it was higher than FIKK7.1 cKO. We found Var2CSA surface expression to be generally variable between parasite lines, which necessitated our conditional approach for gene KO. We believe this is natural variation between parasite lines.

– Figure 1F: This is a nice diagram to convey the organization of the RBC membrane/cytoskeleton and summarize the top human proteins identified by FIKK4.1-T. However, I would recommend color coding the parasite proteins as well – or at least using a different color than the grey that is used for the human proteins that were not enriched by PL since this might confusingly suggest that PfEMP1 and other parasite-encoded knob proteins were not enriched by FIKK4.1-T.

We have coloured the parasite proteins in black to differentiate them from unchanging RBC proteins.

– Figure 2B: Please clarify if these Var2CSA surface presentation assays were carried out in the presence or absence of biotin.

They were performed in the absence of biotin. We have provided additional plots including a 4h biotin pulse to show that this does not impair PfEMP1 trafficking.

– Figure S2A: several major biotinylated bands in the KAHRP-TurboID input are noticeably absent in the avidin pull down; most notable, two major bands at about 80 and 90 kD and a band >135 kD. Curiously, these bands also appear much less prominent in the time course in S2C. Any comment on why this might be?

We assume that the proteins are strongly retained on the beads and are not solubilised, or they precipitate during the incubation phase of pull down. We have not further commented on this here because they are indeed less prominent or absent in the shorter pulses in the timecourse, which is used for all mass-spectrometry experiments. Our interpretation is that these could be proteins of highly abundance along the trafficking pathway as they don’t prominently show up in the pulsed samples. From our data we are not able to identify the nature of the proteins, although it could be interesting to that in the future if they represent intermediate trafficking complexes.

– Line 464: this should be "KO/WT" as in the Y-axis of 5F.

This has been changed.

– Lines 507-514: It is also possible that other FIKKs might function redundantly to support PfEMP1 trafficking.

Correct, statement has been added.

In Figure 3C, the "RBC cytosol" label should be added to the bottom of the PV group as in the "RBC surface" and "Maurer's Clefts" groups to avoid possible confusion about topology. Also, J dots (which should probably be labeled "J dots" instead of just "Dots" in keeping with conventional nomenclature), are thought to be membrane-less organelles and thus the membrane in the diagram may be confusing. Additionally, the authors should clarify where topology has been biochemically determined (I only found this for SBP1 with refs 54 and 55, lines 520-521) vs where it is suggested by their biotinylated peptide data.

We cannot conclude from our data in which orientation the proteins are inserted into the PV, or on which side the biotinylation occurs. We have changed ‘Dots’ to ‘punctate in RBC cytosol’ as the localisation isn’t definitive for these proteins. PTP7 has been moved to the maurer’s cleft category as it is better suited. Many of these proteins are likely trafficked in chaperoned complexes such as the J-dots where the TM domain is not membrane inserted – the schematic depicts their likely topology if/when inserted into a membrane – for example some MC proteins are trafficked to the RBC surface in later stages. This has been clarified in the figure legend. We have clarified that topology has been determined in the literature for PfEMP1, RIFIN, and SBP1.

– Figure 5A: It may be useful to the reader to note in the figure legend that the FIKK9s mutant deletes 7 FIKKS while all others are individual FIKK knockouts.

Yes this has been added.

– Line 926: The DMSO condition is not technically "WT" as it contains a 3xHA-2A-NeoR tag on FIKK4.2.

Changed to ‘intact’.

Reviewer #3 (Recommendations for the authors):The paper is well written, and the data is very clear. There are a few suggestions to improve the work further:1. Some of the figures are very busy and contain many panels. This makes it difficult to sometimes follow.

We have evaluated the figures but believe all the data shown is important for the manuscript and should be included in the main figures – we have used the supplemental figures where possible for less important data. We hope with the implementation of the suggestions from the reviewers and resulting changes makes the manuscript easier to follow.

2. For IFA images shown it would be good to also show light microscopy to clearly show the whole cell as well.

These have been included as supplementary data, and an approximate cell outline is shown on the main figures.

3. As far as I understand disruption of FIKK4.1 leads to var2CSA not being translocated to the surface of the iRBC. It would therefore not assemble into knobs. Also, it means that it would accumulate extensively in the MCs. This raises a number of questions in relation to the interpretation of the data:

It is important to note that deletion of FIKK4.1 leads to a surface translocation defect of ~50% and is NOT completely blocked. Therefore, we don’t expect to see all- or – nothing phenotypes.

a. Are changes observed a result of a true overall change in structure or conformation or rather represent the fact that there is now a gap (due to missing var2CSA) that now creates accessibility.

This is a possibility, but overall the amount of PfEMP1 in the knobs is relatively small so we do not know how much of a gap a reduction of 50% of PfEMP1 would make, and if the proteins would move into that gap. Answering these questions will require single molecule quantification, or PerTurboID experiments with lines where PfEMP1 surface translocation is completely blocked, which should then result in higher accumulation of proteins at the knobs which we could compare to. This is too ambitious for this project, but we anticipate that over the course of the next years we will build much more information and be able to answer that question more definitively.

b. Does the absence of var2CSA on the surface change overall RBC membrane characteristics that now for example allow PSAC to become more mobile and therefore come into closer proximity to the tagged proteins?

This is a possibility: disruption of FIKK4.1 also reduces the rigidity of the RBC, which could lead to higher mobility of the RHOPH complex.

We have added “The reduction in PfEMP1 at the parasite surface and its potential accumulation along the trafficking pathway may have a downstream effect on the mobility and accessibility of surrounding proteins”.

c. Are changes observed in the biotinylation of var2CSA a result of it getting stuck in MCs and if this is the case wouldn't one expect to see this also in the presence of FIKK4.1 as much of PfEMP1 remains in MCs.

We do not know if more PfEMP1 is retained in the Maurer’s clefts, or if PfEMP1 is delivered to the knobs, but is not translocated efficiently. To test this, we would need to generate a Maurer’s cleft sensor line in an FIKK4.1 KO to quantify PfEMP1 levels. We have plans to do this in the future. Clearly, IFAs are not sensitive enough to pick up a difference.

4. In Figure 5. The authors show that FIKK4.1 impacts surface location of PfEMP1 but not RIFIN. To achieve this the authors used tagged LILRB1 and LAIR1 to demonstrate this. However, there are a number of issues that need to be considered.a. In line 434 the authors state that RIFINs are surface exposed and cite the Saito et al., 2017 paper in support. However, as far as I understand the authors do not directly demonstrate that RIFINs are on the surface of the iRBC. They only show binding of the LILRB1 to the surface of the infected RBC. Moreover, the authors show that LILRB1 binding only happens in a small fraction of iRBC and is rapidly lost upon culturing. Can the authors be certain that the signal they detect is indeed due to RIFIN?

We believe the Saito et al. paper allows us to infer that any binding between LILRB1/LAIR1 and iRBC is due to RIFINs. There is more evidence of a direct complex between the RIFINs and LILR1B, which would be hard to explain if the RIFIN would not reach the surface of the iRBC https://www.ncbi.nlm.nih.gov/pmc/articles/PMC7116854/. Uninfected RBC do not bind these ligands, and they showed that transgenic lines expressing specific RIFINs bound to LILRB1 or LAIR1. While it is possible that another parasite protein also binds to these ligands, it is unlikely that our parasite lines don’t express any of the several RIFINs which do bind them. Any changes to RIFIN surface presentation would therefore be observed. We observed about 5-20% binding of iRBC to either LILRB1 or LAIR1, consistent with a mixed population of RIFINs.

b. Since there is only a small subset of RIFIN that bind either LAIR or LILRB1 it would be interesting to note their expression level or whether disruption of FIKK4.1 has any impact on them.

There is no change in RIFIN surface levels so we would not expect expression levels to be changing. Whether other RIFINs may change in expression levels is unknown, but given a lack of selective pressure in our cell culture conditions and that FIKK4.1 is a exported kinase, we do not necessarily expect changes in parasite gene expression levels.

[Editors' note: further revisions were suggested prior to acceptance, as described below.]

The manuscript has been improved and will be acceptable for publication upon a minor revision addressing some remaining issues as outlined below:Reviewer #3 (Recommendations for the authors):Overall, the authors have addressed the issues raised by the reviewers.However, I am still not completely convinced by the arguments of the authors in relation to RIFIN surface expression.I have looked at the presented data carefully again and also had another look at the Harrison et al. paper mentioned.While there is little doubt that some RIFIN interact with LLIRB and LAIR there is no demonstration in any of the papers I am aware of that these RIFIN are actually expressed on the surface of the iRBC. NK cell inactivation can be achieved by secretion via microvessels or other mechanisms that do not involve surface expression. Moreover, the MFI is just around 100 as compared to around 600 for var2CSA expression. This raises some questions on how specific to RIFIN the observed signal really is.I would recommend that the authors take this into consideration and tune down the conclusion a bit.

First the intensity values for LILRB1 and LAIR1 binding are actually very high for individual iRBCs (but not uiRBCs), when considering the histograms in 5G rather than 5H (which was incorrectly labelled). However, there is a bigger spread, reflecting the variability of the expression of rifins within the population. But the ability of the LAIR1 and LILRB1 reagents to recognize the proteins on the surface of unfixed and unpermeabilised iRBCs, but not on uiRBCs, strongly argues for their presence on the surface. The specificity of these reagents has been extensively tested and is supported by structural biology analysis and others have previously shown these to bind to unpermeabilised iRBCs using FACS analysis. While this does not preclude additional secretion, it certainly provides strong evidence for the presence of rifins on the surface.

Second we have generated a PF3D7_1254800 (LILRB1 binding rifin) Turbo-ID line linked to a selection marker to force expression for an unrelated project in the lab. With this cell line, we observe a marked increase of LILRB1 binding to iRBCs, strongly supporting the validity of the argument for this rifin on the surface. This increase is not observed in a cell line that force expresses a var2csa:PfEMP1:TurboID variant, showing the specificity of the signal. This cannot be included in this manuscript as it is completely unrelated, but hopefully convinces the reviewer (and you) that our confidence in our data has merit.

**Author response image 1. sa2fig1:** 

These data do not preclude additional secretion of RIFINs, for example via microvesicles (we presume the reviewer does not mean microvessels), but show that expression on the surface certainly occurs. This is also in line with experiments for other rifins, which has been tested in more detail (for example here) https://malariajournal.biomedcentral.com/articles/10.1186/s12936-015-0784-2.If the reviewer is aware of data that shows secretion of the rifins we tested it woud add a layer to their function, but not invalidate the data of us and others. But we don’t think we need to discuss the possibility of secretion in our paper in the absence of such data in the public domain.